# LEAP: Local ECT-Based Learnable Positional Encodings for Graphs

**Juan Amboage**[1]**, Ernst Röell**[1,2,3]**, Patrick Schnider**[4,5]**, Bastian Rieck**[1,2]

[1]AIDOS Lab, University of Fribourg, Switzerland
[2]Institute of AI for Health, Helmholtz Munich, Germany
[3]Technical University of Munich, Germany
[4]Department of Computer Science, ETH Zurich, Switzerland
[5]Department of Computer Science, University of Basel, Switzerland

## Abstract

Graph neural networks (GNNs) largely rely on the message-passing paradigm, where nodes iteratively aggregate information from their neighbors. Yet, standard message passing neural networks (MPNNs) face well-documented theoretical and practical limitations. Graph positional encoding (PE) has emerged as a promising direction to address these limitations. The Euler Characteristic Transform (ECT) is an efficiently computable geometric–topological invariant that characterizes shapes and graphs. In this work, we combine the differentiable approximation of the ECT (DECT) and its local variant ($\ell$-ECT) to propose LEAP, a new end-to-end trainable local structural PE for graphs. We evaluate our approach on multiple real-world datasets as well as on a synthetic task designed to test its ability to extract topological features. Our results underline the potential of $\ell$-ECT-based encodings as a powerful component for graph representation learning pipelines.

## 1 Introduction

Graphs are the preferred modality in numerous scientific domains, permitting the study of dyadic relationships in an efficient manner. Their broad applicability comes with several challenges that make them harder to process with standard deep learning architectures. Among these characteristics are (i) a mixture of geometrical information (via node and edge features) and topological information (via the edges), (ii) highly variable cardinalities even within the same dataset, and (iii) a lack of a canonical representation. The development of suitable models is thus crucial for advancing the field of *graph representation learning*. Contemporary research largely focuses on *message passing neural networks* (MPNNs), i.e., architectures that are based on local diffusion-like concepts. While powerful, they exhibit intrinsic limitations: For instance, MPNNs tend to lose "signals" in graphs of high diameter (Di Giovanni et al., 2023; Rusch et al., 2023; Zhang et al., 2023), and may be incapable of efficiently leveraging substructure information (Chen et al., 2020).

As an alternative to MPNNs, inspired by the transformer architecture (Vaswani et al., 2017), recent work focuses on *positional encodings* (PEs) and *structural encodings* (SEs) of graphs, denoting functions that assign embeddings to nodes based on locality or relational information, respectively (Dwivedi et al., 2023; Kreuzer et al., 2021; Rampášek et al., 2022). Most PEs/SEs are based on *either* geometrical aspects (like coordinates, curvature, or distances) *or* topological aspects (like Laplacians or random walks), which may limit their practical expressivity. To overcome this, we propose LEAP, a new end-to-end-trainable positional encoding that leverages *both* geometry *and* topology. Being based on a local, learnable variant of the Euler Characteristic Transform (ECT), a geometrical-topological invariant, LEAP is easy to calculate and highly expressive.

Our paper contains the following **contributions**:

1. We propose a new graph positional encoding based on local ECTs, which is highly flexible and permits end-to-end training, specifically geared to work with geometric graphs.
2. We observe that our method captures structural differences in graphs even in case the node features are *non-informative*, thus also permitting to solve learning tasks for non-attributed graphs.
3. We conduct extensive experiments on benchmark datasets that demonstrate that our method yields *improved predictive power* in comparison to existing positional encodings when used in conjunction with graph neural networks.

## 2 BACKGROUND

Before introducing our learnable positional encoding, we provide a short self-contained summary of message-passing, positional encoding in the context of graphs, and the Euler Characteristic Transform.

### 2.1 MESSAGE PASSING

Graph Neural Networks (GNNs) are specifically designed to operate on graph-structured data. A large subclass of GNNs are Message Passing Neural Networks (Gilmer et al., 2017, MPNNs). MPNNs represent each node by a vector that is iteratively updated by aggregating neighboring representations. Hence, the state of a node $v$ at step $t$, denoted $\boldsymbol{h}_v^{(t)}$, is computed as

$$\boldsymbol{h}_v^{(t)} = \text{UPDATE}\left(\boldsymbol{h}_v^{(t-1)}, \text{AGGREGATE}\left(\{\boldsymbol{h}_u^{(t-1)} : u \in \mathcal{N}(v)\}\right)\right), \tag{1}$$

where both AGGREGATE and UPDATE are learnable functions and $\mathcal{N}(v)$ denotes the neighbors of node $v$. Following von Rohrscheidt & Rieck (2025), we refer to a graph $\mathcal{G}$ together with feature vectors for each of its nodes as a *featured graph* and adopt the following notation.

**Definition 1.** *A featured graph is a pair $(\mathcal{G}, x)$, where $\mathcal{G}$ is a (non-directed) graph, and $x$ is a map that assigns each node $v \in V(\mathcal{G})$ a feature vector $x(v) \in \mathbb{R}^d$. We denote the set of nodes of $\mathcal{G}$ by $V(\mathcal{G})$, and the set of edges by $E(\mathcal{G})$.*

Despite their popularity, common MPNNs are limited by phenomena such as oversquashing (Di Giovanni et al., 2023), oversmoothing (Rusch et al., 2023; Zhang et al., 2023), or restricted expressive power (Chen et al., 2020; Xu et al., 2019). Multiple approaches have been proposed to address these challenges, for instance by (i) modifying graph connectivity via virtual nodes (Cai et al., 2023; Grötschla et al., 2024), (ii) combining message passing with global attention (Rampášek et al., 2022), or (iii) imbuing a model with topology-based inductive biases (Horn et al., 2022; Verma et al., 2024).

### 2.2 GRAPH POSITIONAL ENCODINGS

Inspired by positional encodings in transformers (Vaswani et al., 2017), graph positional encodings emerged as a way to inject structural information into the node features. Architectures such as GPS (Rampášek et al., 2022) combine multiple PEs, enabling global-attention layers to incorporate graph structure. Graph PEs have also been shown to benefit standard MPNNs (Dwivedi et al., 2022; 2023; Ma et al., 2021; Verma et al., 2025). Rampášek et al. (2022) propose a categorization of graph PEs into *Positional Encodings* and *Structural Encodings*, further subdivided into *local*, *global*, or *relative* variants. Two commonly-used graph positional encodings are the Random Walk Positional Encoding (Dwivedi et al., 2022, RWPE) and the Laplacian Positional Encoding (Maskey et al., 2022, LaPE), which we will briefly introduce below. Both methods inspired several other non-learnable approaches (Grötschla et al., 2024; Lim et al., 2023; Maskey et al., 2022; Rampášek et al., 2022), as well as learnable ones (Lim et al., 2023, SignNet).

**Random Walk Positional Encoding (RWPE).** For any node $v \in V(\mathcal{G})$, Dwivedi et al. (2022) define the $k$-dimensional RWPE of $v$, denoted by $\boldsymbol{p}_v^{\text{RWPE}_k}$ as

$$\boldsymbol{p}_v^{\text{RWPE}_k} := [\mathbf{RW}_{vv}, (\mathbf{RW})_{vv}^2, \dots, (\mathbf{RW})_{vv}^k] \in \mathbb{R}^k, \tag{2}$$

where $\mathbf{RW} := \boldsymbol{A}(\mathcal{G})\boldsymbol{D}(\mathcal{G})^{-1}$ is the random walk matrix of the graph $\mathcal{G}$, $\boldsymbol{A}(\mathcal{G})$ denotes the *adjacency matrix* of $\mathcal{G}$, and $\boldsymbol{D}(\mathcal{G})$ denotes the *degree matrix* of $\mathcal{G}$. Rampášek et al. (2022) categorize RWPE as a *local structural encoding*.

**Laplacian Positional Encoding (LaPE).** The *normalized Laplacian matrix* of $\mathcal{G}$ is given by $\boldsymbol{L}(\mathcal{G}) = \boldsymbol{I} - \boldsymbol{D}(\mathcal{G})^{-1/2}\boldsymbol{A}(\mathcal{G})\boldsymbol{D}(\mathcal{G})^{-1/2}$, where $\boldsymbol{I}$ denotes the identity matrix. The LaPEs of the nodes in $\mathcal{G}$ are constructed from the eigendecomposition of $\boldsymbol{L}(\mathcal{G}) = \boldsymbol{Q}^{\top}\Lambda\,\boldsymbol{Q}$. Given the eigenvalues sorted in ascending order $\lambda^{(1)}, \ldots, \lambda^{(K)}$, with corresponding eigenvectors $\boldsymbol{q}^{(1)}, \ldots, \boldsymbol{q}^{(K)}$, Dwivedi et al. (2023) define the $k$-dimensional LaPE ($\boldsymbol{p}_v^{\text{LaPE}_k}$) of a node $v$ as

$$\boldsymbol{p}_v^{\text{LaPE}_k} := [\boldsymbol{q}_v^{(i)}, \boldsymbol{q}_v^{(i+1)}, \ldots, \boldsymbol{q}_v^{(i+k-1)}] \in \mathbb{R}^k, \tag{3}$$

where $i$ is the index of the first non-trivial eigenvector. Since LaPE employs the eigendecomposition of the full graph, it is considered to be a *global positional encoding* (Rampášek et al., 2022) .

## 2.3 THE EULER CHARACTERISTIC TRANSFORM (ECT)

The *Euler Characteristic Transform* (ECT) originated as a method to summarize simplicial complexes, i.e., higher-order domains (Turner et al., 2014). Being an expressive and computationally favorable summary statistic, it has found many interesting applications in the biomedical sciences (Amézquita et al., 2021), computer vision (Cisewski-Kehe et al., 2023; Jiang et al., 2020), statistical functional analysis (Crawford et al., 2020), and, more recently, in generative tasks (Röell & Rieck, 2025). For an extensive exposition of the applications and a review of the literature, we refer the reader to Munch (2025) or Rieck (2025).

We will restrict our exposition to the case of graphs, consisting of *vertices* and *edges*. For graphs, the *Euler characteristic* is defined as the number of nodes minus the number of edges; it is a topological invariant of the graph. When the Euler characteristic of two graphs is different, the graphs are not topologically equivalent, permitting us to distinguish them.[1] However, as many graphs have the same Euler characteristic, its expressive power remains limited. By moving to a *multi-scale* variant of the Euler characteristic, we obtain the ECT, which combines geometrical and topological information to obtain a more expressive representation. Specifically, given a featured graph $(\mathcal{G}, x)$, we calculate the inner product of its attributes with a unit vector $\theta \in \mathbb{S}^{d-1}$, referred to as a *direction*, and consider the pre-image of the inner product to obtain a monotonically increasing sequence of subgraphs of $\mathcal{G}$. Tracking the Euler characteristic along that sequence indexed by $t \in \mathbb{R}$ yields the *Euler Characteristic Curve* (ECC) in the direction of $\theta$. The map that sends each direction vector to its corresponding ECC is called the *Euler Characteristic Transform* (ECT). For graphs, it is defined as

$$\text{ECT} \colon \mathbb{S}^{d-1} \times \mathbb{R} \to \mathbb{Z}$$
$$(\theta, t) \mapsto \sum_{v \in V(\mathcal{G})} \mathbb{1}_{[\langle \theta,\, x(v)\rangle,\, \infty)}(t) - \sum_{e \in E(\mathcal{G})} \mathbb{1}_{[\max_{u \in e} \langle \theta,\, x(u)\rangle,\, \infty)}(t). \tag{4}$$

Somewhat surprisingly, given a sufficiently large *finite* number of directions, the ECT is *injective* on geometric graphs and geometric (simplicial) complexes (Curry et al., 2022; Ghrist et al., 2018), i.e., distinct inputs yield distinct ECTs. One limiting factor to the applicability of the ECT in a deep learning setting is the lack of differentiability with respect to the direction vectors and input coordinates. However, by approximating the indicator function of Equation (4) with a sigmoid function, we obtain the *Differentiable Euler Characteristic Transform* (Röell & Rieck, 2024, DECT), which may be integrated into standard deep learning pipelines. This formulation of the ECT provides a *global* summary of a shape, but certain graph learning tasks benefit from a *local* perspective of the graph around a node of interest. As a *static*, i.e., non-trainable, extension to the ECT, the *local Euler Characteristic Transform* (von Rohrscheidt & Rieck, 2025, $\ell$-ECT), constitutes a variant based on local neighborhoods with favorable properties for node classification. Given a featured graph $(\mathcal{G}, x)$ with $x \colon V(\mathcal{G}) \to \mathbb{R}^d$, and a vertex $v$, the local ECT of $v$ with respect to $m \in \mathbb{N}$ is defined as

$$\ell\text{-ECT}_m[\mathcal{G}, x; v] := \text{ECT}[\mathcal{N}_m(v, \mathcal{G}), x|_{V(\mathcal{N}_m(v,\mathcal{G}))}], \tag{5}$$

where $\mathcal{N}_m(v, \mathcal{G})$ denotes a neighborhood of $v$, whose locality is controlled by the hyperparameter $m$. The following result by von Rohrscheidt & Rieck (2025) relates the $\ell$-ECT to MPNNs.

**Theorem 1.** *Let $(\mathcal{G}, x)$ be a featured graph, and let $\{\ell\text{-ECT}_1[\mathcal{G}, x; v]\}_v$ be the set of the 1-hop $\ell$-ECTs of all the vertices $v \in V(G)$. Then $\{\ell\text{-ECT}_1[\mathcal{G}, x; v]\}_v$ provides all the (non-learnable) needed information to perform a single message passing step on $(\mathcal{G}, x)$.*

---

[1]Formally, homotopy-equivalent topological spaces have the same Euler characteristic.

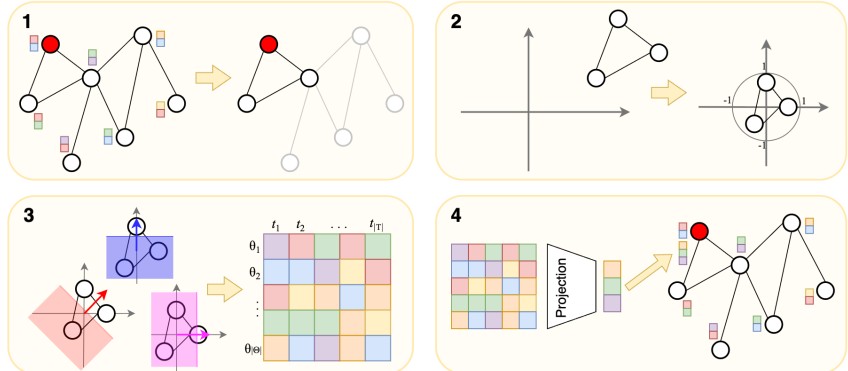

Figure 1: Steps for computing the LEAP PE using 1-hop neighborhoods. (1) The neighborhood of a node in a featured graph is selected. (2) Normalization of the neighborhood features. (3) Computation of the differentiable ECT. (4) Projection of the matrix representation of the ECT to get the PE vector.

The required non-learnable information for a single message passing step refers to the fact that for a given vertex $v$, one can theoretically recover the features of the neighboring nodes from the $\ell$-ECT. This result highlights the power of the 1-hop $\ell$-ECT for graph representation learning. Moreover, von Rohrscheidt & Rieck (2025) show that the $\ell$-ECT is sufficiently expressive to perform subgraph counting, one of the limitations of traditional message passing architectures (Chen et al., 2020). This illustrates that ECT-based methods can be *more powerful* than traditional message passing neural networks in certain cases.

## 3 METHODS

This section introduces the Local ECT and Projection PE (LEAP), a *learnable* local structural graph PE based on the $\ell$-ECT. As part of this encoding, we present strategies for projecting the ECT of a shape into a lower-dimensional space.

### 3.1 $\ell$-ECT BASED POSITIONAL ENCODING

Given a featured graph $(\mathcal{G}, x)$ with $d$-dimensional node features, which may be static (i.e., the original node features or another PE), or learned (i.e., hidden states at some step of an MPNN), let $\mathbb{T} \subset [0, 1]$ be a finite set of thresholds and $\Theta \subset \mathbb{S}^{d-1}$ a finite set of directions. The $k$-dimensional LEAP PE of a node $v \in V(\mathcal{G})$ is constructed as follows:

1. Compute the $m$-hop subgraph $\mathcal{N}_m(v, \mathcal{G})$ around node $v$.
2. Given the set of nodes $\{u_1, \ldots, u_n\} = V(\mathcal{N}_m(v, \mathcal{G}))$, mean-center their feature set $\{x(u_1), \ldots, x(u_n)\}$ and divide each element by the maximum norm in the centered set to obtain new features $\mathbb{F} = \{f(u_1), \ldots, f(u_n)\} \subset \mathbb{S}^{d-1}$, where $f \colon V(\mathcal{N}_m(v, \mathcal{G})) \to \mathbb{F}$ denotes the mapping between each node in the $m$-hop and its normalized feature vector.
3. Compute the matrix $M \in \mathbb{R}^{|\Theta| \times |\mathbb{T}|}$ whose $(i, j)$ entry is the differentiable approximation of the ECT of $(\mathcal{N}_m(v, \mathcal{G}), f)$ at $(\theta_i, t_j) \in \Theta \times \mathbb{T}$.
4. Lastly, a learnable projection $\phi \colon \mathbb{R}^{|\Theta| \times |\mathbb{T}|} \to \mathbb{R}^k$ maps $M$ to a vector $\text{PE}(v) \in \mathbb{R}^k$, which is the final positional encoding of node $v$.

**Remark 1.** LEAP *is* not *a static pre-processing step on the graph. On the contrary, it can be integrated in graph neural network architectures to be trained in an end-to-end fashion.*

The previous remark highlights a key difference between LEAP and graph PEs like LaPE and RWPE. This is also an important distinction from the prior use of the $\ell$-ECT, which was introduced as a static, non-learnable extension of node features, with neighborhood connectivity being disregarded as the ECT was calculated on node neighborhoods as if they were point clouds rather than graphs (von Rohrscheidt & Rieck, 2025). In addition, LEAP permits the set of directions $\Theta$ to be randomly initialized and then *either* kept fixed *or* optimized during training. LEAP can also be applied to

learned graph features and, since it integrates with any GCN, it is naturally applicable to both graph-level and node-level tasks. By contrast, the DECT is geared towards generating graph-level descriptors (Röell & Rieck, 2024).

**Remark 2.** *Within the categorization of Rampášek et al. (2022), LEAP is a* local structural encoding.

The locality of LEAP comes from computing each node's encoding only from its $m$-hop subgraph. Thus, locality is controlled by the hop number $m$, which serves as a hyperparameter. By default, we suggest 1-hop neighborhoods, making our method *as scalable* as message passing, but we also describe two ways to control the locality of LEAP:

- Use a larger hop number $m$. While straightforward, it should be noted that two nodes may differ in their $m$-hop neighborhoods while becoming identical at $(m + 1)$-hops.[2]
- Alternatively, we compute LEAP multiple times for each node with increasing $m$, then concatenate the results to obtain a PE that captures how the $m$-hop neighborhoods evolve as $m$ grows.

We also note that two nodes in a graph that share identical $m$-hop neighborhoods receive the same LEAP PE, since the second step in the computation of the PE yields identical outputs. This aligns directly with the definition of *local structural encoding* given in Rampášek et al. (2022, Table 1). Moreover, consider a node whose normalized $m$-hop neighborhood features form a geometric graph embedding.[3] If we could access the ECT of that subgraph rather than an approximation, then by the injectivity results of the ECT (Curry et al., 2022; Ghrist et al., 2018) we would have all the information required to recover the neighborhood's structure.[4]

## 3.2 ECT PROJECTION STRATEGIES

Since LEAP aims to capture structural information, it should be invariant to scaling and rotations of neighborhood features. Step 2 above addresses normalization, but to minimize the effect of rotations, the projection in Step 4 should be *permutation invariant* with respect to the ECCs. However, this requirement is often ignored in practice (Röell & Rieck, 2024). In the remainder of this section, we present five projection strategies for LEAP, some of which explicitly enforce this invariance.

**Linear projection:** We "flatten" the $\ell$-ECT of each node into a vector $\boldsymbol{v} \in \mathbb{R}^D$ with $D = |\Theta| \cdot |\mathbb{T}|$, following Amézquita et al. (2021). We then apply a linear projection by multiplying $\boldsymbol{v}$ with a learnable matrix $\boldsymbol{W} \in \mathbb{R}^{k \times D}$. This projection is *not* permutation invariant with respect to the ECC, and the number of learnable parameters with respect to $|\Theta|$ and $|\mathbb{T}|$ is $\mathcal{O}(|\Theta| \cdot |\mathbb{T}|)$.

**One-dimensional convolutions:** We treat the $\ell$-ECT of each node as a multichannel time series, where thresholds act as time steps and each ECC defines a channel. Several 1D convolutions are concatenated, and the resulting channels are averaged to produce a vector that is used as an input to an MLP. This projection is *not* permutation invariant with respect to the order of the directions, and the number of learnable parameters with respect to $|\Theta|$ and $|\mathbb{T}|$ is $\mathcal{O}(|\Theta| + |\mathbb{T}|)$.

**DeepSets:** We treat the $\ell$-ECT of a node as a set of $|\mathbb{T}|$-dimensional vectors, corresponding to the ECCs along different directions in $\Theta$, processing this set using an architecture inspired by DeepSets (Zaheer et al., 2017): Given the set of vectors corresponding to the ECCs we have $\text{PE} = \text{MLP}_2(\sum_{\theta \in |\Theta|} \text{MLP}_1(\text{ECC}_\theta))$. This projection strategy is permutation invariant wrt. the directions of the ECT, and its number of learnable parameters is independent of $|\Theta|$.

**Attention:** We treat the $\ell$-ECT of a node as a set of $|\mathbb{T}|$-dimensional vectors, corresponding to the ECCs along the different directions in $\Theta$, and process this set by a transformer encoder with a single attention head. To obtain the PE, we apply an MLP to the sum of the generated ECC representations. Due to the use of a self-attention without any positional encoding, the projection is permutation invariant, and the number of learnable parameters depends on $|\mathbb{T}|$ but not on $|\Theta|$.

**Attention with PE:** As a variant of the previous projection, instead of feeding the transformer encoder the set of ECCs directly, we concatenate each $\text{ECC}_\theta$ with the corresponding direction $\theta \in \Theta$ before passing it to the encoder. This yields a permutation invariant projection strategy, while incorporating information about the directions along which the ECCs were computed.

---

[2]For sufficiently large $m$, this strategy yields identical PEs for all nodes within the same connected component.
[3]In general, there is no guarantee this will occur.
[4]We design an experiment to test the ability of LEAP to capture topological features of a graph, see Section 4.1.

### 3.3 PROPERTIES

We first discuss the *computational complexity* of our method. Given an $m$-hop subgraph $\mathcal{N}_m(v, \mathcal{G})$ for each vertex $v$, calculating the $\ell$-ECT has a total computational complexity of $\mathcal{O}(\sum_v |\mathcal{N}_m(v, \mathcal{G})|)$. In the worst case, each subgraph is the *complete* graph on $n$ vertices, leading to an overall complexity of $\mathcal{O}(n^3)$. For *sparse graphs* whose $m$-hop neighborhood is of the order of $m = \mathcal{O}(n)$, we obtain a worst-case complexity of $\mathcal{O}(n^2)$. Finally, assuming *bounded degree*, this reduces to a worst-case complexity of $\mathcal{O}(n)$, which is asymptotically equal to one step of message passing. Moreover, individual $\ell$-ECTs can be computed *in parallel*. In terms of expressivity, von Rohrscheidt & Rieck (2025) provide the theoretical foundation for our work, stating that, given a sufficiently large number of directions, the injectivity of the $\ell$-ECT guarantees that it is *more* expressive than message passing. However, we consider the main contribution of our work to be the development of a novel local positional encoding and its empirical evaluation, in the spirit of Rampášek et al. (2022), leaving a more in-depth theoretical analysis for future work.

## 4 EXPERIMENTS

We conduct experiments to evaluate different aspects of LEAP, investigating (i) its ability to capture structural properties *independent* of node features, (ii) its impact on the performance of different graph neural network architectures and the effect of learning the directions of the transform, (iii) its performance on a large-scale dataset with 202,579 graphs (Chen et al., 2019), (iv) its behavior when applied to learned node features in the *HIV* dataset (Wu et al., 2018), and (v) the effect of hyperparameters. Subsequently, *LEAP-L* indicates that the directions for LEAP were randomly initialized and learned during training, while *LEAP-F* denotes that the directions remained fixed.

### 4.1 SYNTHETIC DATASET

We introduce a synthetic dataset of 40,000 graphs to test whether LEAP can capture structural differences *independent* of node features, thus proving that LEAP is indeed a *structural encoding*. Each graph has three nodes and contains either zero, one, two, or three edges, yielding a classification task with four classes based on edge count. The node features are uniformly sampled from the unit disk $D_1 \subset \mathbb{R}^2$ to make the task purely structural. We use a standard GCN and GAT architectures as base models, and compare them to the same model enhanced with LEAP added as structural positional encoding. For the computation of the ECT used in LEAP, we use 16 directions with a resolution of 16, summarizing each graph into a $16 \times 16$ ECT. The models enhanced with LEAP achieve a perfect accuracy of $100.0 \pm 0.0$, demonstrating LEAP's ability to capture structural properties *independent* of the node features. By contrast, the GCN and GAT models alone exhibited lower accuracies ($71.83 \pm 0.27$ and $69.44 \pm 0.82$, respectively), demonstrating their inability to capture relevant structural graph properties when informative node features are not available[5].

### 4.2 CLASSIFYING REAL-WORLD DATASETS

Table 1: Best approach (architecture, PE strategy, and projection strategy) and relative accuracy improvement with respect to the worst performing baseline for TU classification datasets. In all cases the best result was achieved using our PE strategy.

| DATASET | BEST METHOD | WORST | BEST | GAIN (%) |
|---|---|---|---|---|
| LETTER-H | NoMP + LEAP-L+ 1D Conv | 41.6 | 81.6 | 96.2 |
| LETTER-M | NoMP + LEAP-L+ 1D Conv | 57.8 | 88.5 | 53.1 |
| LETTER-L | NoMP + LEAP-L+ 1D Conv | 80.4 | 98.0 | 21.9 |
| FINGERPRINT | NoMP + LEAP-L+ Linear | 48.8 | 55.7 | 14.1 |
| COX2 | GAT + LEAP-L+ Attn w/ PE | 77.7 | 80.1 | 3.1 |
| BZR | NoMP + LEAP-L+ Linear | 78.3 | 84.7 | 8.2 |
| DHFR | GCN + LEAP-L+ Attn w/ PE | 70.1 | 77.6 | 10.7 |

We evaluate LEAP on several graph classification datasets from the TU benchmark (Morris et al., 2020). Our aim is to evaluate (i) the capacity of LEAP to enhance existing graph neural networks with structural information, (ii) compare LEAP with existing PEs, and (iii) investigate in which architecture LEAP induces the largest increase in accuracy. Of particular interest is the evaluation on the *Alchemy_full* (Chen et al., 2019) dataset, as the regression targets are rotation invariant with respect to the node features. For this dataset, we normalize the regression targets so that all 12 tasks are on the

---

[5]Although LaPE and RWPE are computed independently of node features, we also performed this experiment using these PEs. Like LEAP, they achieved perfect accuracy.

same scale. For the *HIV* dataset (Wu et al., 2018), where nodes have categorical features, we exploit the end-to-end differentiability of the ECT by extending the architecture with a learnable embedding layer that maps these features into $\mathbb{R}^3$, where LEAP is computed.

**Architectures.** We fix four "backbone" architectures to which we add different positional encodings, namely (i) a GCN (Kipf & Welling, 2017), (ii) a GAT (Veličković et al., 2018), (iii) a GIN (Xu et al., 2019), and (iv) NoMP ("no message passing"), a model that we introduce based on a transformer encoder. Following Maggs et al. (2024), we use five message-passing layers and 32-dimensional hidden states for GCN and GAT. For the *Alchemy* dataset, we scale the architectures to 10 layers with 64-dimensional hidden states. The NoMP architecture projects node features into a 16-dimensional latent space with a linear layer, followed by a single self-attention layer that produces a 16-dimensional state for each node. The final classification is performed by a feedforward layer. We chose hyperparameters to match the parameter count of GCN/GAT. By design, NoMP ignores graph structure unless given positional encodings, thus permitting us to evaluate the ability of each PE to encode relevant structural properties. Finally, as positional encodings, we consider the following baselines: (i) No positional encoding, (ii) RWPE, which, like LEAP, is a local structural PE under the categorization of Rampášek et al. (2022), making it a particularly relevant baseline, and (iii) LaPE, a widely used graph PE that, unlike LEAP, captures global positional information.

**Experimental setup.** All experiments use 5-fold cross-validation and are trained with the Adam optimizer for up to 100 epochs with early stopping enabled. As a loss term, we use the *cross entropy loss* except for the *Alchemy* dataset, where we use the mean squared error loss. We use 10-dimensional PEs for all types and datasets. The only difference between the backbones with or without a PE is that the input dimension of the backbone increases by 10 when a PE is used. The Euler Characteristic Transform in LEAP is calculated with 16 directions and 16 thresholds. To simplify the setup, we keep all hyperparameters of LEAP's projection strategies *fixed* across all datasets. Despite this, as we describe below, we observe high predictive performance across a variety of datasets.

## 4.3 RESULTS

Table 2: Surprisingly, increasing the neighborhood size ($\mathcal{N}_m$) does not improve the efficacy of LEAP, showing that the 1-hop neighborhood is sufficient.

| METHOD | $\mathcal{N}_m$ | LETTER-H | LETTER-M | LETTER-L |
|---|---|---|---|---|
| LEAP-F | 1 | 81.29 ± 1.91 | 88.00 ± 1.89 | 96.27 ± 0.84 |
| | 2 | 74.44 ± 3.26 | 84.31 ± 0.76 | 94.13 ± 1.05 |
| | 1, 2 | 77.91 ± 1.82 | 84.76 ± 1.40 | 96.09 ± 0.34 |
| LEAP-L | 1 | 80.62 ± 3.58 | 86.49 ± 2.20 | 96.18 ± 1.24 |
| | 2 | 72.76 ± 2.66 | 85.11 ± 1.29 | 93.38 ± 0.84 |
| | 1, 2 | 78.13 ± 2.83 | 85.96 ± 1.42 | 95.64 ± 1.51 |

Table 4 reports the results for LEAP and the baselines in combination with different architectures across the various classification datasets. For every dataset–architecture combination, the two LEAP variants (F/L) achieve the best and second-best performance. When combined with GCN and NoMP architectures, *learning* the directions of LEAP consistently improves performance in comparison to keeping them fixed. For GAT, learning the directions slightly reduces performance compared to the fixed variant of LEAP

in 3 of the 7 datasets. For all datasets, the overall best-performing architecture–PE combination always uses LEAP with learned directions.

Table 1 reports the relative increase of the best-performing method compared to the worst-performing method. We observe the largest gains from using LEAP on the *Letter* and *Fingerprint* datasets. For the datasets *COX2*, *BZR*, and *DHFR*, the advantage over the baselines is less pronounced, likely due to their smaller size, making it harder to benefit from richer features. We observe the largest improvement on *DHFR*, the largest dataset among the three. Table 4 also shows that NoMP *without* positional encodings outperforms baseline GNNs on the *Letter-High* and *Letter-Low* datasets, highlighting the limitations of MPNNs, i.e., models *without* structural information may achieve better results.

We also evaluate LEAP on the *Alchemy* and *HIV* datasets from the TU and MoleculeNet benchmarks, respectively, following the same setup described above. The *Alchemy* dataset is significantly larger compared to the other datasets allowing LEAP to extract more meaningful information from the data. For the *Alchemy* dataset, we show the $R^2$ score and for the *HIV* dataset, we report AUROC, due to large class imbalances. Figure 2 shows the results for LEAP with the GCN backbone and the various PE methods, showing a clear advantage for LEAP (with learned directions) over all baselines. For

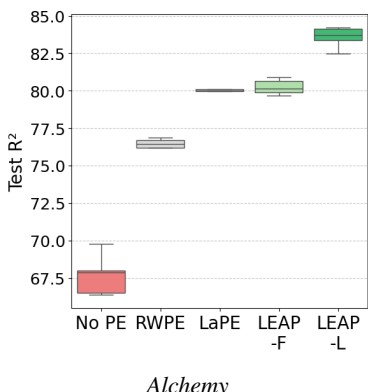 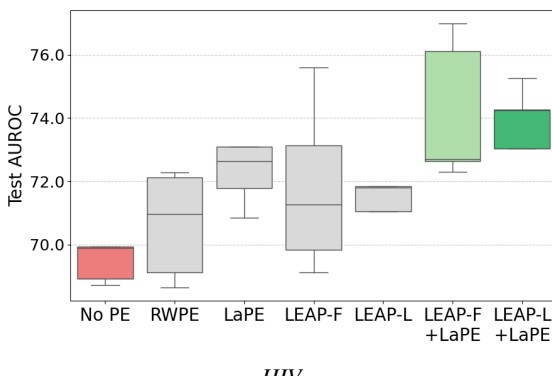

*Alchemy*                  *HIV*

Figure 2: Results for different PE strategies on the *Alchemy* and *HIV* datasets reporting the $R^2$ and AUROC using a GCN. Colors rank the PEs from **best**, second best to worst. LEAP with learnable direction significantly outperforms other methods on the *Alchemy* dataset while performing second best on the *HIV* dataset.

the *HIV* dataset, results exhibit a large degree of variability, and this is the only case where neither of the two LEAP variant (L/F) outperforms all baselines. Still, both variants surpass RWPE, which is in the same category of *local structural encodings*. LaPE yields the strongest performance for a single PE, suggesting that *global positional* information, which cannot be captured by LEAP, may be particularly relevant in this case. For this reason, we also evaluate the use of LaPE and LEAP combined through concatenation. We observe that the combination of both PEs yields the best overall results. This showcases how these two PE strategies can successfully complement each other. Please refer to Figure S.1 and Figure S.2 for the full results.

## 4.4 ABLATIONS

After having established that LEAP captures essential structural information to be used with multiple graph neural network architectures, we further aim to investigate the sensitivity of LEAP with respect to its various components. In particular we hope to further understand how (i) the choice of projection method, (ii) the number of hops, and (iii) the embedding dimension impact the performance of LEAP.

Table 3: Ablation for the embedding dimension of the projection. LEAP is stable wrt. the dimension, performing consistently well.

| EMB. | PE | LETTER-H | LETTER-M | LETTER-L |
|---|---|---|---|---|
| 2 | LaPE | 66.31 ± 1.20 | 94.71 ± 1.29 | 76.76 ± 1.95 |
| | RWPE | 73.82 ± 2.00 | 94.62 ± 0.51 | 83.47 ± 1.68 |
| | LEAP-F | 81.16 ± 1.66 | 96.27 ± 1.47 | 86.76 ± 1.86 |
| | LEAP-L | 78.53 ± 3.30 | 95.56 ± 0.31 | 87.38 ± 0.99 |
| 5 | LaPE | 64.44 ± 3.54 | 94.22 ± 0.61 | 82.09 ± 1.64 |
| | RWPE | 75.64 ± 1.18 | 94.67 ± 0.96 | 85.82 ± 0.76 |
| | LEAP-F | 79.78 ± 0.57 | 96.98 ± 0.25 | 86.76 ± 1.11 |
| | LEAP-L | 80.40 ± 1.41 | 96.44 ± 0.59 | 88.36 ± 0.64 |
| 10 | LaPE | 65.02 ± 1.58 | 91.11 ± 2.23 | 76.93 ± 2.52 |
| | RWPE | 79.24 ± 1.43 | 94.67 ± 0.97 | 84.53 ± 1.32 |
| | LEAP-F | 80.13 ± 2.04 | 96.68 ± 0.78 | 86.59 ± 2.01 |
| | LEAP-L | 80.68 ± 2.42 | 96.99 ± 1.05 | 87.24 ± 2.31 |
| 20 | LaPE | 64.80 ± 2.49 | 93.24 ± 0.96 | 77.64 ± 2.12 |
| | RWPE | 76.76 ± 1.36 | 95.38 ± 1.12 | 86.49 ± 1.21 |
| | LEAP-F | 80.84 ± 1.89 | 95.56 ± 0.65 | 87.42 ± 0.75 |
| | LEAP-L | 79.91 ± 1.53 | 96.40 ± 0.79 | 86.89 ± 1.56 |

**Projection strategies.** We repeated all experiments using the five proposed LEAP projection strategies such that each projection strategy has approximately similar small parameter budgets, comprising 1K–5K parameters. Table S.1 reports the results; and we find *no* single projection consistently outperformed the others, showing the best projection to be dependent on the dataset–architecture combination. However, a remarkable fact is that learnable directions did *on average* outperform the fixed set of directions, underpinning the benefits of learnable directions as compared to using them as static features.

**Locality parameter sensitivity.** We also study the effect of the locality parameter in LEAP by repeating the experiments on the *Letter* datasets, originally performed with 1-hop neighborhoods, using instead 2-hop neighborhoods and the concatenation of LEAP embeddings from 1- and 2-hop neighborhoods.[6] For this

---

[6]In the concatenation setting, each embedding is computed with half the target dimension so that the final representation matches the dimension of the other approaches.

Table 4: Accuracy results for different PE strategies when using a variety of GNN architectures for multiple datasets from the TU Dataset benchmark. We evaluate the GCN, GAT and GIN architectures with and without LEAP. Additionally we evaluate LEAP on a non-message passing architecture (NoMP). Best results are **green**, second best are green, and worst are red. For every dataset, our approach achieves the best and second best results.

| MODEL | PE | COX2 | BZR | DHFR | LETTER-H | LETTER-M | LETTER-L | FINGERPRINT |
|---|---|---|---|---|---|---|---|---|
| GCN | No PE | 77.9 ± 1.0 | 81.9 ± 3.3 | 71.6 ± 1.4 | 41.6 ± 4.1 | 63.5 ± 2.0 | 80.4 ± 1.0 | 48.8 ± 1.4 |
|  | RWPE | 78.4 ± 0.5 | 79.5 ± 2.2 | 73.0 ± 2.4 | 60.9 ± 1.7 | 68.9 ± 2.7 | 83.2 ± 1.4 | 49.4 ± 0.6 |
|  | LaPE | 78.4 ± 0.9 | 80.3 ± 1.2 | 70.4 ± 2.8 | 55.3 ± 2.6 | 75.8 ± 2.6 | 89.2 ± 1.2 | 48.1 ± 1.8 |
|  | LEAP-F | 79.2 ± 0.6 | 82.5 ± 2.4 | 74.1 ± 5.2 | 72.2 ± 3.3 | 82.6 ± 1.4 | 95.8 ± 1.1 | 55.6 ± 1.1 |
|  | LEAP-L | 79.4 ± 1.0 | 82.5 ± 1.6 | 77.6 ± 2.8 | 74.2 ± 1.5 | 83.6 ± 1.3 | 96.0 ± 0.9 | 55.1 ± 1.2 |
| GAT | No PE | 78.2 ± 0.6 | 80.5 ± 2.0 | 73.7 ± 1.8 | 41.9 ± 3.2 | 58.4 ± 3.7 | 89.4 ± 0.7 | 50.5 ± 0.6 |
|  | RWPE | 79.0 ± 1.4 | 78.3 ± 1.1 | 70.9 ± 2.4 | 63.0 ± 3.0 | 69.0 ± 1.8 | 90.8 ± 1.5 | 50.4 ± 0.8 |
|  | LaPE | 77.9 ± 1.0 | 80.3 ± 1.2 | 70.4 ± 2.7 | 54.7 ± 5.3 | 75.2 ± 2.3 | 89.6 ± 1.5 | 48.9 ± 1.0 |
|  | LEAP-F | 79.2 ± 1.6 | 82.0 ± 3.2 | 75.7 ± 3.0 | 70.2 ± 2.2 | 83.2 ± 1.1 | 95.8 ± 0.8 | 55.1 ± 0.6 |
|  | LEAP-L | 80.1 ± 2.2 | 83.7 ± 2.9 | 76.5 ± 3.8 | 73.5 ± 2.1 | 82.4 ± 1.6 | 95.2 ± 0.9 | 54.9 ± 0.7 |
| GIN | No PE | 78.2 ± 0.5 | 79.5 ± 1.4 | 69.3 ± 4.8 | 47.7 ± 0.8 | 65.0 ± 3.9 | 82.7 ± 1.5 | 48.4 ± 1.4 |
|  | RWPE | 78.6 ± 1.6 | 79.3 ± 1.0 | 72.0 ± 4.9 | 54.4 ± 2.3 | 64.9 ± 3.7 | 81.6 ± 2.4 | 50.0 ± 1.6 |
|  | LaPE | 78.2 ± 0.5 | 79.5 ± 1.7 | 61.0 ± 0.1 | 55.0 ± 3.5 | 75.2 ± 3.5 | 84.4 ± 3.5 | 49.8 ± 1.9 |
|  | LEAP-F | 79.0 ± 0.9 | 81.2 ± 1.4 | 73.9 ± 4.1 | 60.2 ± 4.8 | 76.3 ± 2.0 | 93.3 ± 1.1 | 54.4 ± 1.1 |
|  | LEAP-L | 79.6 ± 1.6 | 81.0 ± 1.4 | 76.2 ± 3.2 | 62.7 ± 3.4 | 77.6 ± 1.7 | 94.0 ± 1.9 | 55.3 ± 1.4 |
| NoMP | No PE | 77.9 ± 0.8 | 79.8 ± 2.6 | 70.1 ± 3.4 | 63.4 ± 1.0 | 57.8 ± 0.9 | 89.7 ± 1.3 | 50.7 ± 0.5 |
|  | RWPE | 77.7 ± 1.3 | 80.9 ± 1.7 | 73.3 ± 1.5 | 79.2 ± 1.4 | 84.5 ± 1.3 | 94.7 ± 1.0 | 51.3 ± 0.7 |
|  | LaPE | 77.7 ± 1.0 | 81.2 ± 3.2 | 70.5 ± 3.5 | 65.0 ± 1.6 | 76.9 ± 2.5 | 91.1 ± 2.2 | 50.5 ± 1.2 |
|  | LEAP-F | 79.0 ± 0.6 | 83.2 ± 1.7 | 74.3 ± 6.1 | 81.3 ± 1.9 | 88.0 ± 1.9 | 97.2 ± 0.3 | 55.7 ± 1.1 |
|  | LEAP-L | 78.6 ± 0.8 | 84.7 ± 2.7 | 75.7 ± 2.7 | 81.6 ± 1.9 | 88.5 ± 2.5 | 98.0 ± 0.4 | 56.3 ± 1.4 |

ablation, we use the NoMP model so that the models can only access structural information through the PE, and we use *attention with PE* as the LEAP projection strategy. Table 2 shows that the 1-hop neighborhood yields the best performance across all datasets, followed by the concatenated 1- and 2-hop version, respectively.

**Impact of PE dimension.** To better understand the effect of increasing the embedding dimension for the projection strategies, we vary the size of the embedding dimension on the *Letter* datasets. The original experiment was ran with PE dimension 10 for both LEAP and the baselines, and we now repeat it with the embedding dimension set to $\{2, 5, 10, 20\}$, respectively. As before, we fix the architecture to NoMP so that models access structural information only through the PE, and use *attention with PE* as the projection strategy for LEAP. The results in Table 3 show that across *all* evaluated PE dimensions and datasets, LEAP outperforms both RWPE and LaPE.

**DECT hyperparameters and comparison.** We assessed LEAP's sensitivity to the DECT hyperparameters by varying the number of directions in $\{2, 4, 8, 16, 32\}$ and smoothing parameter in $\{2, 4, 8, 16, 32, 64, 128\}$. LEAP remained robust, outperforming baselines across all settings; see Figure S.5. Finally, Table S.2 summarizes the comparison of LEAP with DECT for graph classification tasks. LEAP outperforms two variants of the DECT (with different parameter budgets) on most datasets, which further underscores the utility of learnable directions.

## 5 CONCLUSION AND FUTURE WORK

We presented LEAP, a new *learnable local structural positional encoding* for graphs based on the $\ell$-ECT. To the best of our knowledge, this is the *first* approach to integrate the $\ell$-ECT into deep learning architectures in an end-to-end trainable fashion.

Our experiments show that LEAP consistently outperforms established baselines across multiple architectures and datasets, with learned directions further improving performance in most tasks, thereby highlighting the benefits of making this step trainable. Additionally, we introduced a synthetic task in which our approach achieved perfect accuracy, demonstrating its ability to capture

topological information independent of node features, which the evaluated MPNNs (GCN/GAT) alone failed to recover. Taken together, these results highlight the potential of $\ell$-ECT encodings for *topological deep learning* (Papamarkou et al., 2024) and graph representation learning tasks. LEAP is particularly well-suited to provide local structural information to architectures that rely on global attention mechanisms, where graph structure is not directly modeled and multiple PEs are combined to capture complementary notions of graph position.

**Limitations.** While LEAP provides a learnable way to capture local structural information it has some limitations. First, it is not a purely structural PE, as it requires *node features* to compute the ECTs. However, these features can be *learned*, and in the synthetic dataset, our approach succeeded even though the features were irrelevant to the prediction targets. Second, LEAP relies on a differentiable approximation of the discretized ECT applied to normalized $m$-hop subgraphs, which are not necessarily geometric, so the theoretical guarantees of the *exact* ECT (e.g., injectivity) may not fully carry over; we expect this to be interesting for future work. Finally, unlike other graph PEs, such as LaPE or RWPE, where the only hyperparameter is the embedding dimension, LEAP introduces several hyperparameters (among others, a smoothing parameter of the ECT approximation, the number of directions, and the number of discretization steps). In practice, however, we fixed these across datasets and architectures, nevertheless observing consistently strong performance. Our ablation studies further serve to demonstrate the robustness to these choices.

**Future work.** We envision several directions for future research. First, drawing on prior work (von Rohrscheidt & Rieck, 2025), we aim to formalize the theoretical expressivity of LEAP, noting that theoretical expressivity and empirical performance are often not correlated. Combining LEAP with positional encodings that capture complementary aspects of graph structure and embedding it within more sophisticated architectures may further improve performance and expressivity. Another promising line of work is to make the ECT step fully differentiable. Instead of discretizing along a fixed grid, treating thresholds as trainable parameters would allow the model to focus on informative regions and optimize their positions jointly with the other parameters. Using learned features, we also plan on assessing the performance of LEAP on non-attributed graph datasets, i.e., datasets that are fully structural. Finally, since the ECT can be applied to higher-order datasets (Ballester et al., 2025) comprising, for instance, simplicial complexes or cell complexes (Hoppe et al., 2025), we believe that LEAP could be extended to this modality, thus serving as a generalizable addition to the topological deep learning toolbox.

## REPRODUCIBILITY STATEMENT

Our code can be accessed at https://github.com/aidos-lab/LEAP. All experiments used a fixed seed; the full configurations can be found in the `experiments` folder.

## ACKNOWLEDGMENTS

*In loving memory of Teresa Paz Camps.*

First and foremost, facing a turbulent decision-making process, we want to thank the reviewers and the AC for believing in the merits of our work. We are particularly grateful for the support by reviewer f41T, who continued to champion our work throughout the revision process. This work has received funding from the Swiss State Secretariat for Education, Research, and Innovation (SERI). B.R. wishes to dedicate this paper to his daughter Aurélie.

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

## Appendix (Supplementary Materials)

## A    RESULTS FOR THE ALCHEMY AND HIV DATASETS

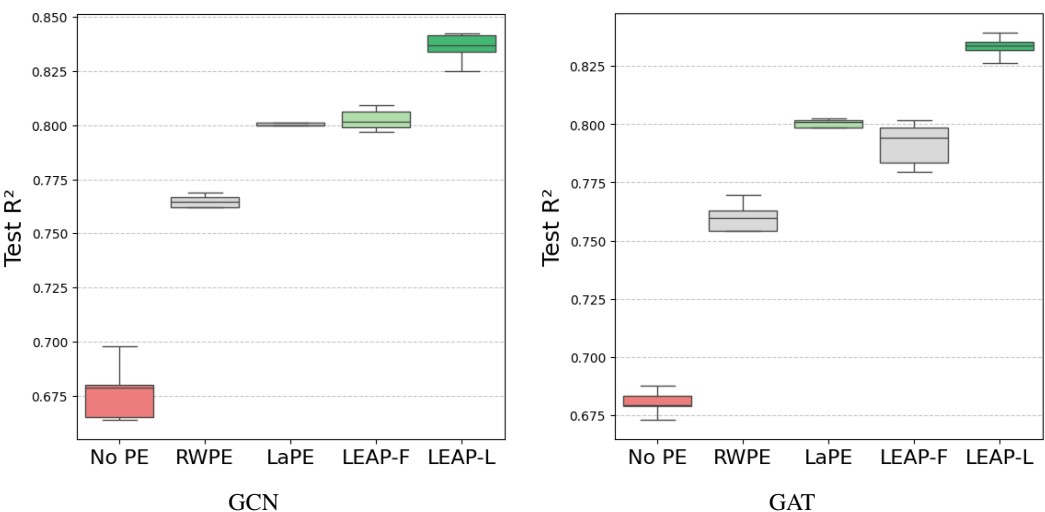

Figure S.1: $R^2$ results for different PE strategies on the *Alchemy* dataset using the GCN and GAT architectures. Best results in terms of mean $R^2$ are **green**, second best are green, and worst are red.

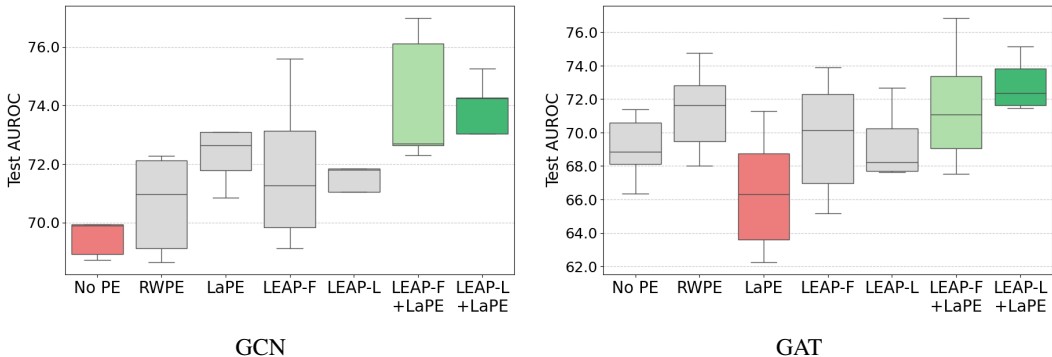

Figure S.2: AUROC results for different PE strategies on the *HIV* dataset using the GCN and GAT architectures, respectively. Best results in terms of mean AUROC are **green**, second best are green, and worst are red.

## B  VALIDATION METRICS

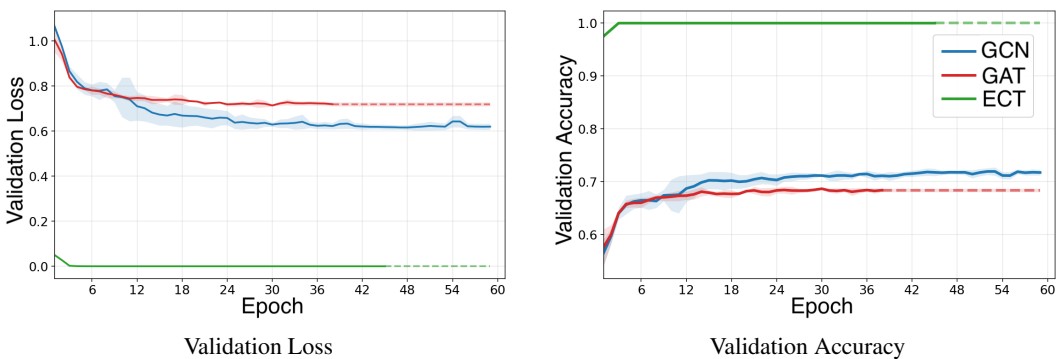

Validation Loss                    Validation Accuracy

Figure S.3: Validation loss and accuracy per training epoch for the synthetic dataset for the baseline GCN, GAT, and LEAP. Our method achieves a perfect score in both metrics and convergence immediately. The shadows indicate one standard deviation over 5 runs and the dashed line means that model training finished earlier because of early stopping.

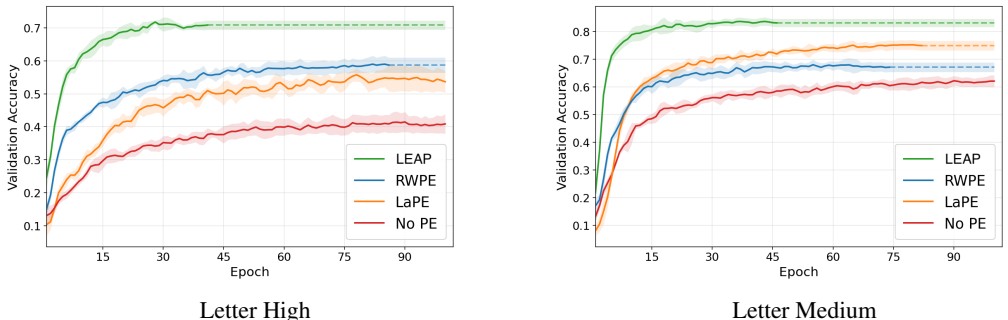

Letter High                         Letter Medium

Figure S.4: Validation accuracy per training epoch for the Letter High (left) and Letter Medium (right) datasets for different PE strategies using a GCN architecture. Our method achieves the best results and converges faster. The shadows around the curves indicate the standard deviation over 5 runs and the dashed line means that training ended due to early stopping.

# C   ADDITIONAL EXPERIMENTS

Table S.1: Accuracy results for all real-world datasets when varying the strategy of the LEAP PE with fixed and learnable directions for different models. The backbone architectures have around 4K parameters and we show the additional parameters each positional encoding introduces.

| | | | COX2 | | BZR | | DHFR | | LETTER-H | | LETTER-M | | LETTER-L | | FINGERPRINT | |
|---|---|---|---|---|---|---|---|---|---|---|---|---|---|---|---|---|
| MODEL | PROJ. METHOD | PARAMETERS | LEAP-F | LEAP-L | LEAP-F | LEAP-L | LEAP-F | LEAP-L | LEAP-F | LEAP-L | LEAP-F | LEAP-L | LEAP-F | LEAP-L | LEAP-F | LEAP-L |
| GCN | Linear | 4K+2.5K | 79.2±0.6 | 79.4±1.0 | 78.8±0.6 | 82.5±2.0 | 70.9±3.2 | 74.9±4.0 | 72.2±3.3 | 74.2±1.5 | 82.8±1.4 | 83.6±1.3 | 95.2±0.9 | 96.0±0.9 | 55.6±1.1 | 54.7±1.5 |
| | Attn | 4K+5K | 78.4±1.3 | 79.0±0.9 | 81.7±2.8 | 82.5±1.6 | 74.1±5.2 | 77.3±4.1 | 59.8±2.8 | 62.7±2.6 | 74.4±1.5 | 73.9±4.6 | 92.3±1.3 | 94.5±1.4 | 53.0±1.9 | 54.4±1.2 |
| | Attn PE | 4K+5K | 78.2±1.2 | 78.8±1.3 | 82.5±2.4 | 82.5±3.1 | 73.2±3.7 | 77.6±2.8 | 67.2±1.5 | 68.6±1.8 | 82.0±0.8 | 82.9±2.2 | 95.8±1.1 | 94.7±1.6 | 54.1±1.3 | 55.1±1.2 |
| | DeepSets | 4K+.5K | 78.2±0.6 | 79.0±1.2 | 79.0±1.2 | 81.7±3.5 | 71.2±2.6 | 73.3±3.6 | 59.2±1.9 | 63.4±2.1 | 72.4±0.6 | 76.0±1.1 | 91.4±1.8 | 92.0±0.6 | 52.3±1.3 | 54.1±1.3 |
| | 1D Conv | 4K+.9K | 78.0±1.2 | 79.2±2.0 | 79.8±1.9 | 82.5±2.7 | 71.7±1.2 | 76.7±3.4 | 66.4±2.8 | 63.1±3.2 | 81.6±1.5 | 81.7±3.5 | 94.2±1.5 | 93.4±1.4 | 55.6±1.1 | 54.0±2.2 |
| GAT | Linear | 4K+2.5K | 78.4±1.2 | 79.7±2.0 | 79.3±1.8 | 81.7±2.4 | 75.7±3.0 | 76.3±2.2 | 70.2±2.2 | 73.5±2.1 | 82.4±2.3 | 82.4±1.6 | 95.8±0.8 | 95.1±0.9 | 55.1±0.6 | 54.8±2.1 |
| | Attn | 4K+5K | 78.4±0.5 | 78.8±0.8 | 82.0±3.4 | 82.2±2.1 | 74.9±3.1 | 73.3±2.8 | 62.0±1.2 | 62.9±3.1 | 75.0±2.1 | 79.7±1.4 | 94.4±0.1 | 93.4±1.0 | 52.0±1.3 | 53.5±1.5 |
| | Attn PE | 4K+5K | 78.8±0.8 | 80.1±2.2 | 82.0±3.2 | 79.5±0.7 | 73.2±3.1 | 75.9±5.8 | 66.7±2.0 | 65.0±3.5 | 83.2±1.1 | 79.3±2.4 | 94.0±1.3 | 95.1±1.1 | 54.5±1.8 | 54.5±1.8 |
| | DeepSets | 4K+.5K | 79.2±1.6 | 79.7±1.8 | 81.2±2.0 | 82.7±4.7 | 71.2±4.1 | 76.5±3.8 | 58.8±2.6 | 56.4±4.3 | 75.4±3.1 | 76.6±2.3 | 94.6±1.4 | 93.1±1.3 | 52.9±2.2 | 51.0±1.1 |
| | 1D Conv | 4K+.9K | 78.4±0.9 | 78.6±1.7 | 81.7±4.0 | 83.7±2.9 | 70.6±2.3 | 75.7±1.5 | 67.3±2.0 | 68.1±1.3 | 80.9±1.3 | 80.5±2.8 | 93.6±2.0 | 95.2±0.9 | 54.8±1.5 | 54.9±0.7 |
| GIN | Linear | 4K+2.5K | 78.4±0.8 | 77.9±2.2 | 79.3±1.4 | 78.8±0.6 | 76.2±3.2 | 70.6±3.2 | 62.7±3.4 | 60.2±4.8 | 77.6±1.7 | 74.7±3.6 | 94.0±1.9 | 93.3±1.1 | 55.0±1.2 | 54.4±1.1 |
| | Attn | 4K+5K | 78.4±0.8 | 78.2±0.5 | 78.8±0.6 | 78.5±1.9 | 61.0±0.1 | 73.9±4.1 | 53.6±4.0 | 46.3±8.1 | 73.2±5.2 | 65.2±9.4 | 92.8±1.3 | 89.6±0.8 | 55.3±1.4 | 53.9±0.9 |
| | Attn PE | 4K+5K | 78.2±0.5 | 78.2±0.5 | 78.8±0.6 | 78.8±0.6 | 61.0±0.1 | 61.0±0.1 | 58.4±4.8 | 52.8±6.9 | 75.1±3.5 | 76.3±2.0 | 92.8±1.0 | 90.6±6.0 | 54.0±1.4 | 53.7±0.5 |
| | DeepSets | 4K+.5K | 79.6±1.6 | 79.0±0.9 | 81.0±1.4 | 80.2±2.0 | 74.7±3.4 | 69.2±2.2 | 49.5±3.5 | 53.6±6.9 | 71.2±2.0 | 67.4±4.6 | 89.5±1.9 | 87.2±3.8 | 54.5±0.7 | 53.3±0.5 |
| | 1D Conv | 4K+.9K | 78.2±0.5 | 78.4±1.4 | 79.0±1.2 | 81.2±1.4 | 72.1±5.6 | 70.1±6.8 | 57.9±4.7 | 52.7±2.7 | 76.5±3.7 | 72.7±6.3 | 91.8±1.7 | 90.6±0.8 | 53.8±0.4 | 54.2±1.1 |
| NoMP | Linear | 4K+2.5K | 79.0±0.6 | 78.6±0.8 | 83.2±1.7 | 84.7±2.7 | 74.3±6.1 | 74.9±3.3 | 79.5±1.2 | 79.4±1.1 | 86.0±2.2 | 85.4±1.5 | 96.7±0.8 | 96.4±0.7 | 55.7±1.1 | 56.3±1.4 |
| | Attn | 4K+5K | 78.2±0.5 | 78.2±0.5 | 81.7±2.8 | 79.0±0.9 | 68.3±5.7 | 71.7±4.1 | 79.0±1.8 | 81.3±1.9 | 84.8±0.9 | 86.5±2.6 | 96.1±0.6 | 97.2±0.8 | 53.8±0.7 | 54.8±1.2 |
| | Attn PE | 4K+5K | 78.4±0.4 | 77.7±1.5 | 78.8±0.6 | 78.8±0.6 | 64.2±4.8 | 72.1±3.7 | 81.3±1.9 | 80.6±3.6 | 88.0±1.9 | 86.5±2.2 | 96.3±0.8 | 96.2±1.2 | 54.8±1.4 | 55.3±1.1 |
| | DeepSets | 4K+.5K | 78.4±0.8 | 78.0±0.5 | 83.2±2.1 | 82.0±2.8 | 69.5±3.0 | 72.6±3.6 | 78.0±2.0 | 79.2±1.9 | 86.0±2.2 | 87.5±2.0 | 96.3±0.6 | 96.6±0.8 | 54.0±0.4 | 54.3±1.0 |
| | 1D Conv | 4K+.9K | 78.0±1.9 | 78.1±1.1 | 81.0±1.7 | 79.3±1.8 | 71.6±3.2 | 75.7±2.7 | 81.1±0.9 | 81.6±1.9 | 87.0±2.0 | 88.5±2.5 | 97.2±0.3 | 98.0±0.4 | 54.5±1.0 | 54.1±0.4 |

Figure S.5: We assess the sensitivity of LEAP with respect to the hyperparameters used in the ECT. Top row shows the effect of changing the hyperparameters for LEAP-F (fixed directions) and the bottom row for LEAP-L (learnable directions). LEAP consistently outperforms baselines across all settings and is thus robust with respect to the hyperparameters.

Table S.2: We provide a comparison with DECT (Röell & Rieck, 2024). DECT summarizes the graph with a single global ECT and subsequently applies a convolutional neural network for the classification. We compare our method with two variants of DECT, one with 4K parameters and one with 65K parameters. The parameter count in LEAP ranges from 1K to 5K and therefore the comparison with DECT (4K) would be the most appropriate, although we outperform both variants on most datasets.

| MODEL | COX2 | BZR | DHFR | LETTER-H | LETTER-M | LETTER-L |
|---|---|---|---|---|---|---|
| DECT (4K) | 70.4 ± 0.9 | 81.8 ± 3.2 | 67.9 ± 5.0 | 63.8 ± 6.0 | 76.2 ± 4.8 | 91.5 ± 2.1 |
| DECT (65K) | 74.6 ± 4.5 | 84.3 ± 6.1 | 72.9 ± 1.6 | 85.4 ± 1.3 | 86.3 ± 2.0 | 96.8 ± 1.2 |
| LEAP-L (GCN) | 79.4 ± 1.0 | 82.5 ± 1.6 | 77.6 ± 2.8 | 74.2 ± 1.5 | 83.6 ± 1.3 | 96.0 ± 0.9 |
| LEAP-L (GAT) | 80.1 ± 2.2 | 83.7 ± 2.9 | 76.5 ± 3.8 | 73.5 ± 2.1 | 82.4 ± 1.6 | 95.2 ± 0.9 |
| LEAP-L (NOMP) | 78.6 ± 0.8 | 84.7 ± 2.7 | 75.7 ± 2.7 | 81.6 ± 1.9 | 88.5 ± 2.5 | 98.0 ± 0.4 |

# D    COMPUTATIONAL PERFORMANCE

Table S.3: We report the average training time for the Letter High dataset. The average training time measures the total training time, including potential early stopping. For fair comparison between the various methods we also report the average training time per epoch. LEAP yields both high performance while remaining fast to train.

| METHOD | TRAINING TIME | AVERAGE PER EPOCH |
|--------|---------------|-------------------|
| LaPE   | 60.35         | 0.64              |
| NoPE   | 44.60         | 0.51              |
| RWPE   | 35.77         | 0.50              |
| LEAP-L | 32.51         | 0.93              |
| LEAP-F | 30.08         | 0.84              |

Table S.4: For the Roman Empire dataset (Platonov et al., 2023) the average training time per epoch is reported as well as the initial preprocessing time. Since this dataset consists in a single graph with 22662 nodes, 32927 edges, and 300-dimensional features, we choose it to evaluate how the runtimes of the PEs scale for larger graphs. We report preprocessing time as for this graph size it is *not* negligible. Moreover, in this case, for the LEAP variants, the preprocessing consists in precomputing the node neighborhoods. LaPE requires the eigen decomposition of the full graph, leading to large initial preprocessing times. LEAP remains computationally efficient.

| METHOD | TRAINING TIME PER EPOCH | PREPROCESSING TIME |
|--------|-------------------------|--------------------|
| NoPE   | 0.09                    | 0.01               |
| LaPE   | 0.10                    | 164.83             |
| RWPE   | 0.10                    | 0.18               |
| LEAP-F | 0.58                    | 11.57              |
| LEAP-L | 1.13                    | 11.57              |

