# OpenReview forum: "LEAP: Local ECT-Based Learnable Positional Encodings for Graphs"
_ICLR.cc/2026/Conference — ICLR 2026 Poster_

### Official Review · Reviewer_f41T · 2025-10-24

**Soundness:** 3
**Presentation:** 4
**Contribution:** 2
**Rating:** 6
**Confidence:** 4

**Summary:**

Building upon the local Euler Characteristic Transform ($\ell$-ECT), this paper introduces a new class of learnable *local strucutral encodings* (LSE) that can be integrated in any GNN architecture. To be precise, the main novelty is that such structural encodings are made learnable, since they were already adopted as static descriptors in another work, although very recent. Another novelty lies into a new "projection" layer in the proposed neural architecture, allowing one to reduce a matrix containing the differentiable approximations of $\ell$-ECT, one per vertex, into a final vector: the positional encoding for that vertex. The approach is extremely versatile and the experimental section clearly shows that i) the proposed LSE allows standard GNNs to improve their performance in a variety of tasks and ii) learning the LSEs further increase the improvement and iii)  the choice of the hyper-parameters (e.g. number of directions, time grids) in not constraining, meaning that different choices still lead to good results, except for the hop's ray, in this case the smaller the better.

**Strengths:**

This paper is really well written, clear and concise.  The experimental setup is solid and the results are impressive (more comments later), however I would say the contribution is just "fair" since local and global ECT descriptors already existed. However, I think that this paper has the potential to be accepted at ICLR, so I am **temporarily** ranking it with a 6, but I am ready to raise my note to 8 after the discussion, provided some key points are discussed/addressed.

**Weaknesses:**

A few points deserves to be discussed and/or presented differently. One major drawback, discussed below, and this is the only reason why I do not immediately rate this paper with a 8.

**Questions:**

Major remarks:

i) It is unfortunate that the word "projection" is used in two different contexts: former concerning the definition of ECT, Eq. (4), latter concerning the $\phi$ map at l.198. Moreover, this word plays a role in your acronym and it is quite clear that you refer to the second projection ($\phi$) there but a few words about it would be appreciated.

ii) At line 225 you introduce the notion of *valid graph embedding*. It would be wise to recall the reader what it means.

iii) The experiments of the synthetic dataset are the only ones not being very convincing. I mean: what happens when you provide GCN/GAT with LaPE or RWPE? Isn't the prediction score equal to 1? Even if it were the case it would not be an issue, in my view, since in the rest of the section, by the way, you beat the state-of-the-art.

iv) Here's the weakest point of the paper in my view. There is a number of message passing architectures known to be more expressive w.r.t. GCNs. The first and maybe most important I can think of is graphs isomorphism networks (GINs, [1]). Including them in your Table 4 would turn your paper into a master  piece.

v) At l. 355 I read: "An interesting observation from Table 4 ... ". I think you were too modest in your commentary! What I see in Table 4 is that on 5 columns over 7 the best accuracy score is obtained by a neural architecture **not** being a GNN, provided that it is equipped with LEAP-L. Did about Alchemy and HIV datasets? Did you test NoMP on them?

Minor remarks:

i) At line 239, you need a space between *to* and $|\\Theta |$
ii) Line 248: *rojection* -> *projection*

---

> ### Author Response · Authors · 2025-11-18
> **Response 1/2**
>
> **We would like to thank the reviewer very much for the positive review and their attention to detail in the evaluation of our work\! It is nice to see this enthusiasm, which is much appreciated and highly motivational\!**
>
> The strong experimental results form the main cornerstone of this work and thereby complementing the theoretical foundation laid by \[1\]. In the evaluation, we were also pleasantly surprised that the results were as strong as they were and in combination with the computational performance, believe that this is useful contribution to the topological deep learning literature.
>
> **Please find our responses and comments below;we would very much appreciate further discussion\!**
>
> > The experimental setup is solid and the results are impressive…
>
> Thank you very much\! We have indeed done our best to ensure thorough evaluation in our experiments.
>
> > It is unfortunate that the word "projection" is used in two different contexts: former concerning the definition of ECT, Eq. (4), latter concerning the  map at l.198. Moreover, this word plays a role in your acronym and it is quite clear that you refer to the second projection () there but a few words about it would be appreciated.
>
> Thank you very much for the attention to detail and helping us improve the clarity of the paper. **We have revised the manuscript and rephrased the definition of the ECT in Eq. 4 to prevent the use of the word “projection.”** The term “projection” will now only refer to the projection method of the ECT in LEAP. We believe this to increase clarity of the overall paper and appreciate the suggestion.
>
> >  At line 225 you introduce the notion of *valid graph embedding*. It would be wise to remind the reader what it means.
>
> With a valid graph embedding, we wanted to denote a geometric graph with node features. Thus, in practice, all graphs we consider are embedded graphs in this sense.  **The manuscript has been updated to reflect the context better.** Thanks for the great catch\!
>
> > The experiments of the synthetic dataset are the only ones not being very convincing. I mean: what happens when you provide GCN/GAT with LaPE or RWPE? Isn't the prediction score equal to 1? Even if it were the case it would not be an issue, in my view, since in the rest of the section, by the way, you beat the state-of-the-art.
>
> **We thank the reviewer for the valuable feedback on the synthetic experiments.** Our initial motivation has been to show that LEAP can capture **structural information** even when informative node features are **not present (thus giving credibility to LEAP being a *structural encoding*)**. In addition to evaluating LEAP,  we have now evaluated the GNN architectures with LaPE and RWPE. In both cases the model returns perfect accuracy, **showing that all positional encodings** can capture the graph structure when node features are uninformative. We will update the discussion (potentially in the appendix) for the final version of our manuscript.
>
> Please find the remaining comment in the second response.
>
> —
>
> \[1\] [Diss-l-ECT: Dissecting Graph Data with Local Euler Characteristic Transforms](https://arxiv.org/abs/2410.02622)

---

> ### Author Response · Authors · 2025-11-18
> **Response 2/2**
>
> > Here's the weakest point of the paper in my view. There is a number of message passing architectures known to be more expressive w.r.t. GCNs. The first and maybe most important I can think of is graphs isomorphism networks (GINs, \[1\]). Including them in your Table 4 would turn your paper into a masterpiece.
>
> Thanks for this suggestion! We fully agree that comparing to a wider range of architectures improves the quality of our work and **have evaluated the GIN architecture in our framework**. The results (Table 4 and Table 5\)  are **updated in the revised manuscript** and for reference added below. We find that with the GIN architecture, LEAP performs **very well and consistently improves classification accuracy**.
>
> (Based on the suggestion of Reviewer `8cdq`, we have also added an evaluation with LEAP complemented with a global positional encoding showing the complementary information to be very beneficial for the HIV dataset. )
>
> |                             | COX2                          | BZR                          | DHFR                          | Letter-low                          | Letter-med                          | Letter-high                          | Fingerprint                          |
> | :-------------------------- | :---------------------------- | :--------------------------- | :---------------------------- | :---------------------------------- | :---------------------------------- | :----------------------------------- | :----------------------------------- |
> | NoPE                        | 78.2 $\pm$ 0.5                  | 79.5 $\pm$ 1.4                 | 69.3 $\pm$ 4.8                  | 82.7 $\pm$ 1.5                        | 65.0 $\pm$ 3.9                        | 47.7 $\pm$ 0.8                         | 48.4 $\pm$ 1.4                         |
> | LaPE                        | 78.2 $\pm$ 0.5                  | 79.5 $\pm$ 1.7                 | 61.0 $\pm$ 0.1                  | 84.4 $\pm$ 3.5                        | 75.2 $\pm$ 3.5                        | 55.0 $\pm$ 3.5                         | 49.8 $\pm$ 1.9                         |
> | RWPE                        | 78.6 $\pm$ 1.6                  | 79.3 $\pm$ 1.0                 | 72.0 $\pm$ 4.9                  | 81.6 $\pm$ 2.4                        | 64.9 $\pm$ 3.7                        | 54.4 $\pm$ 2.3                         | 50.0 $\pm$ 1.6                         |
> | LEAP-F                      | 79.0 $\pm$ 0.9                  | 81.2 $\pm$ 1.4                 | 73.9 $\pm$ 4.1                  | 93.3 $\pm$ 1.1                        | 76.3 $\pm$ 2.0                        | 60.2 $\pm$ 4.8                         | 54.4 $\pm$ 1.1                         |
> | LEAP-L                      | 79.6 $\pm$ 1.6                  | 81.0 $\pm$ 1.4                 | 76.2 $\pm$ 3.2                  | 94.0 $\pm$ 1.9                        | 77.6 $\pm$ 1.7                        | 62.7 $\pm$ 3.4                         | 55.3 $\pm$ 1.4                         |
>
> > At l. 355 I read: "An interesting observation from Table 4 ... ". I think you were too modest in your commentary\! What I see in Table 4 is that on 5 columns over 7 the best accuracy score is obtained by a neural architecture not being a GNN, provided that it is equipped with LEAP-L. Did about Alchemy and HIV datasets? Did you test NoMP on them?
>
> Thank you very much for the encouraging words\! **We have updated the text with a firmer claim\!**
>
> We also provide the results below for the `Alchemy` and `HIV` dataset without message passing, extending Figure 3\. For `Alchemy`, **comparing the results with the GNN architectures in Figure 3 in the appendix, the NoMP+LEAP-L ($R^2 of \~0.85$) architecture outperforms both the GCN+LEAP-L ($R^2 of \~0.83$) and GAT+LEAP-L ($R^2 of \~0.83$):**
>
> | Method | $R^2$ Score|
> |:-----------------|:--------------|
> | NoPE | 0.62 $\pm$ 0.0  |
> | LaPE  | 0.72 $\pm$ 0.01 |
> | RWPE  | 0.76 $\pm$ 0.0  |
> | LEAP-F  | 0.79 $\pm$ 0.0  |
> | LEAP-L  | 0.85 $\pm$ 0.0  |
>
> For `HIV`, by contrast, message passing appears to be a **crucial component** to reach high AUROC, and we observe overall worse results than reported (**see also updated Figure 3 in the appendix**) in the main manuscript:
>
> | Method       |AUROC   |
> |:-------|:--------------|
> | LaPE   | 0.7 $\pm$ 0.02  |
> | NoPE   | 0.62 $\pm$ 0.01 |
> | LEAP-F | 0.65 $\pm$ 0.03 |
> | LEAP-L | 0.61 $\pm$ 0.04 |
> | RWPE.  | 0.7 $\pm$ 0.02  |
>
> > Minor remarks.
>
> - At line 239, you need a space between to and
> - Line 248: rojection \-\> projection
>
> Thank you very much for the attention to detail\! We care a lot about presenting a well-written paper for the reviews and appreciate the feedback. **We have fixed this in our revision.**
>
> —
>
> \[1\] [Diss-l-ECT: Dissecting Graph Data with Local Euler Characteristic Transforms](https://arxiv.org/abs/2410.02622)

---

> > ### Comment · Reviewer_f41T · 2025-11-19
> >
> > I wish to thank the authors for answering all of my questions. The additional results they showed (also in answers to other reviewers), reinforce my belief that LEAP has the potential to become a crucial tool in graph representation learning. As promised, I will raise my rating to 8.

---

> > > ### Author Response · Authors · 2025-11-19
> > >
> > > We are very grateful for your support! 🙌 If you have any additional questions, please let us know! We are working further on improving the manuscript in the meantime.

---

### Official Review · Reviewer_8cdq · 2025-10-30

**Soundness:** 2
**Presentation:** 3
**Contribution:** 2
**Rating:** 4
**Confidence:** 3

**Summary:**

This paper proposes LEAP, a learnable graph positional encoding method based on a local Euler Characteristic Transform (ℓ-ECT). The key idea is to compute differentiable, direction-dependent topology-aware signatures localized around each node and incorporate them as positional embeddings in standard GNN architectures. By leveraging both geometric and topological structure, LEAP aims to address limitations of message passing GNNs that struggle to capture higher-order structural patterns. The authors evaluate the method on synthetic and real-world graph classification datasets, comparing to standard positional encodings such as RWPE and Laplacian-based LaPE. Results indicate that LEAP improves performance in many settings, particularly when used with architectures that are otherwise limited in structural awareness.

**Strengths:**

1. The use of local, differentiable Euler characteristic transforms as graph positional encoding is an interesting new direction that meaningfully blends geometric and topological features.
2. The paper clearly explains the limitations of pure message passing networks and why positional encoding is useful.
3. The learnable direction variants of LEAP show thoughtful design toward practical applicability.

**Weaknesses:**

1. While LEAP is motivated as addressing expressivity limits of MPNNs, the paper does not provide formal analysis or guarantees regarding what structural distinctions LEAP can or cannot capture. The argument remains primarily empirical.
2. While LEAP performs well in several TU datasets, its benefits diminish or disappear on datasets where global structure is more important (e.g., HIV), where LaPE performs best. This suggests that LEAP may be inherently limited in capturing global structural contexts, which should be more explicitly acknowledged or explored.

**Questions:**

1. Since LEAP is intrinsically local, have you considered augmenting it with a complementary global encoding? Could LEAP be combined with LaPE without redundancy?
2. What is the computational overhead per graph and per layer when incorporating LEAP? How does it scale with graph size and number of directions?

---

> ### Author Response · Authors · 2025-11-18
> **Response 1/2**
>
> We appreciate the reviewer for recognizing that our method can capture both the geometric features as well as the structural properties of the graph. Practical applicability has also been one of the core considerations and motivation for the extensive evaluation.  Please find our responses below.
>
> > While LEAP is motivated as addressing expressivity limits of MPNNs, the paper does not provide formal analysis or guarantees regarding what structural distinctions LEAP can or cannot capture. The argument remains primarily empirical.
>
> Our paper indeed has a strong focus on the integration of the local ECT into a machine learning pipeline. The theoretical focus of prior work \[1\] permits us to consider the practical application as novel **differentiable structural encoding** (unlike the prior "static" feature augmentation), thus complementing existing work with a new method and its strong evaluation. From a theoretical perspective, prior work shows that the local ECT is theoretically **at least as expressive as message passing** \[1\], all the while capturing additional structural information. That being said, we fully understand that the current version of the manuscript has underilluminated this perspective and **we have updated the manuscript to reflect the work that has been done towards the expressivity of the ECT (Section 3.3)**.
>
> > While LEAP performs well in several TU datasets, its benefits diminish or disappear on datasets where global structure is more important (e.g., HIV), where LaPE performs best. This suggests that LEAP may be inherently limited in capturing global structural contexts, which should be more explicitly acknowledged or explored.
>
> Thank you for raising this excellent point! That the HIV dataset requires a global perspective is something we had not considered; we primarily consider our method to be strong when **local information is useful**. However, to **complement the local perspective with the global** perspective, it is possible to append the embedding of the global ECT to the final representation. Another option would be to directly concatenate the global ECT with the local ECT and project the result.  Overall, we find that our method  provides a flexible framework that can deal with a variety of machine learning problems in a flexible and computationally performant way. **We have now updated the manuscript to discuss these limitations and Section 5** and we will run additional local–global experiments during the rebuttal time. Please also see the **answer below** for a combination of global and local encodings.
>
> Please find the responses to the remaining points below.

---

> ### Author Response · Authors · 2025-11-18
> **Response 2/2**
>
> > Since LEAP is intrinsically local, have you considered augmenting it with a complementary global encoding? Could LEAP be combined with LaPE without redundancy?
>
> Extending the answer above, it is indeed possible to **further enhance LEAP with the global positional encoding of the graph**. We have added LaPE to the LEAP-L to provide this complementary perspective and it results in a **significant increase in accuracy on the HIV** dataset. **The result shows that this not only improves the results, it actually outperforms both methods.**
>
> | Model         | Accuracy     |
> | ---            |  --- |
> | NoPE  | 69.48 $\pm$ 0.60 |
> | LaPE  | 72.94 $\pm$ 2.09 |
> | RWPE   | 70.62 $\pm$ 1.68 |
> | LEAP-F        | 71.79 $\pm$ 2.62 |
> | LEAP-L        | 71.38 $\pm$ 1.72 |
> | LEAP-F & LaPE | 74.14 $\pm$ 2.22 |
> | LEAP-L & LaPE | 73.59 $\pm$ 1.57 |
>
> **We will add these results in the appendix or the main text for the final version of our paper.**
>
> > What is the computational overhead per graph and per layer when incorporating LEAP? How does it scale with graph size and number of directions?
>
> As shown in \[2\], the computation of the ECT is **hardware accelerable and efficient**. During training, the most computationally expensive step is the computation of the $1$- or $2$-hop neighborhoods. As the neighborhoods do not change during training, it is an excellent candidate for preprocessing, further accelerating the training. **We opted not to do so in our experiments, as training times were already reasonable with the recomputation**, and we preferred providing a proof-of-concept implementation. **We have added a discussion on the computational complexity in Section 3.3 to highlight the scalability of our method**. For reference, we provide the training times for the `Letter-High` dataset, indicating that LEAP is **efficient to train in practice**.
>
> | Method | Average Training (s) | Average Time/Epoch (s) | Average Epochs |
> | ---    | ---                  | ---                    | ---            |
> | NoPE   | 44.60            | 0.51               | 86.4           |
> | LaPE   | 60.35            | 0.64               | 93.0           |
> | RWPE   | 35.77            | 0.50               | 71.2           |
> | LEAP-L | 32.51            | 0.93               | 34.8           |
> | LEAP-F | 30.08            | 0.84               | 35.8           |
>
> ---
>
> \[1\] [Diss-l-ECT: Dissecting Graph Data with Local Euler Characteristic Transforms](https://openreview.net/forum?id=GAmmzu6GYS)
> \[2\] [Differentiable Euler Characteristic Transforms for Shape Classification](https://arxiv.org/abs/2310.07630)

---

> > ### Author Response · Authors · 2025-11-24
> >
> > We would like to thank the reviewer again for their efforts in reviewing our
> > work and hope to have addressed their main concerns. We provide **additional
> > context and provided the theoretical motivation**,  **evaluated LEAP with global
> > positional encoding** and **provide a thorough analysis of the runtime and complexity
> > of our method**.
> > The changes in the manuscript are **highlighted in green** and avalable for the reviewers.
> >
> > If the reviewer believes we have adequately addressed their concerns we truly
> > appreciate it if they would reconsider their score and if they have any
> > questions we would be very happy to engage in further discussion.

---

> > > ### Author Response · Authors · 2025-11-28
> > >
> > > We thank the reviewer for their valuable comments. Since the rebuttal is coming to an end (and NeurIPS is about to start), we were wondering if there were any other points that we could clarify or experiments we could run. We are particularly happy about the suggestions for the extended discussion on global versus local aspects of the data (see replies). We believe that we have addressed your main concerns with our rebuttal answers and the revision; if so, we would kindly ask you to reconsider your overall score.

---

### Official Review · Reviewer_6i1u · 2025-11-01

**Soundness:** 3
**Presentation:** 3
**Contribution:** 2
**Rating:** 4
**Confidence:** 2

**Summary:**

The paper proposes LEAP, a learnable positional encoding for graphs based on the local Euler Characteristic Transform ($l$-ECT), which integrates both geometric and topological information and supports end-to-end training. The authors validate LEAP on synthetic tasks and multiple real-world graph datasets, demonstrating its ability to capture structural information even when node features are non-informative.

**Strengths:**

(1) LEAP is the first work to integrate the local ECT into GNNs in an end-to-end trainable manner, combining geometric and topological insights.

(2) The authors evaluate their method on synthetic data, small molecules, image graphs, and large-scale quantum chemistry datasets, showing consistent improvements. The synthetic task clearly demonstrates LEAP’s ability to classify graphs based solely on structural information, highlighting limitations of standard MPNNs.

(3) The paper includes thorough ablation studies on various aspects (Projection strategies, Locality, PE dimension, and DECT hyperparameters) of LEAP, strengthening its conclusions.

**Weaknesses:**

(1) As the authors acknowledge in the limitations, LEAP is not a purely structural PE. It requires node features $x(v)$ to compute the ECT. While the synthetic experiment shows LEAP works even with uninformative features, it is unclear how, or if, LEAP would be applied to graphs without any node features (e.g., graphs represented only by an adjacency matrix).

(2) LEAP introduces more hyperparameters (e.g., number of directions, smoothing, discretization steps) compared to LaPE or RWPE.

(3) Although not analyzed in depth, computing ECT for m-hop subgraphs for every node could be expensive for large graphs.

**Questions:**

The ablation study (Table 2 ) shows that 1-hop neighborhoods perform better than 2-hop or a 1,2-hop combination. This seems counter-intuitive to the MPNN notion that larger receptive fields are better. Why is 1-hop sufficient?

How does LEAP perform on very large graphs?

---

> ### Author Response · Authors · 2025-11-18
> **Response 1/2**
>
> We thank the reviewer for their efforts in the thorough review. We believe the combination of both a geometric perspective combined with a topological perspective to be beneficial in graph learning tasks. **Providing a thorough evaluation of our method has been an important goal and we appreciate the recognition**. The experiments on the synthetic datasets serve to show that LEAP has the ability to capture structural information, even when informative node features are not present (we now **added an explanation in the paper**). In line with the discussion with the other reviewers, we have decided to **extend it and evaluate LaPE and RWPE as well**.
>
> With respect to the points raised by the reviewer, please find our responses below.
>
> > Applying LEAP to graphs without node features
>
> Our primary focus lies on datasets that have node features (e.g., class features or spatial coordinates). Notably, MPNNs require node features. Our recommendation would be to start with random features, or add relevant features through feature engineering (such as curvature, node degree, or centrality measures). As our experiments in Section 4.1 demonstrate, our method still encodes the local graph structure, permitting key structural properties to be captured, even when node features are randomized. We thank the reviewer for this feedback and **have updated our discussion section with more details (which, subject to additional pages for a potential final version, we will expand on)**.
>
> > Hyperparameters of LEAP
>
> LEAP indeed has several hyperparameters that can be chosen and modified depending on the dataset and task. However, we find that our proposed pipeline is **robust with respect to the choice of hyperparameters**, thus providing added value over existing encodings. As we demonstrate in our experiments, choosing $16$ directions with a spatial resolution of $16$ (following the experimental setup of \[1\]) is a robust starting point. For the projection method, we observe that either the attention-based or linear projection perform well across a wide variety of datasets.
>
> When the model has the tendency to over- or underfit, decreasing (resp. increasing) the number of directions and resolution is a suitable option, as well. In choosing the right scale for the sigmoid approximation in the DECT, we follow \[1\] and the settings therein. In a recent paper \[2\], the authors perform an ablation with respect to the scale and derive the principle that setting the scale to $¼$ of the resolution yields the best result. **We will provide additional guidelines on choosing these hyperparameters for the final version of our manuscript**.
>
> Please find the responses to your remaining questions below.

---

> ### Author Response · Authors · 2025-11-18
> **Response 2/2**
>
> > How does LEAP perform on very large graphs?
>
> Thanks for this suggestion. We also believe that computational considerations are an important aspect and **have already updated our revision with a dedicated section on computational complexity**. Calculating the $m$-hop neighborhood might indeed scale unfavorably with the graph size, although it will only have to be done once. Moreover, the calculation of the ECT is an intrinsically parallel operation which can be accelerated on hardware. It should also be noted that global positional positional encodings (such as LaPE) require an eigendecomposition of the adjacency matrix (albeit, remaining static, such a computation is only required once), whereas our calculations are much lighter (but happen per epoch).
>
> For reference, we provide the training time of our current experimental setup for the various types of positional encodings. While our method incurs some additional computational cost, it is still a performant method, and we want to highlight that we did not exhaust all options for better implementations. **Moreover, our current implementation already outperforms existing positional encodings**.
>
> | Method | Average Training (s) | Average Time/Epoch (s) | Average Epochs |
> | ---    | ---                  | ---                    | ---            |
> | NoPE   | 44.60            | 0.51               | 86.4           |
> | LaPE   | 60.35            | 0.64               | 93.0           |
> | RWPE   | 35.77            | 0.50               | 71.2           |
> | LEAP-L | 32.51            | 0.93               | 34.8           |
> | LEAP-F | 30.08            | 0.84               | 35.8           |
>
> Even in the case of large graphs, our method performs well. Below we present the training times for the `Roman-Empire` dataset, showing **fast training and loading (i.e., pre-processing) times**.
>
> | Method | Avg Epoch Time (s) | Avg pre-processing time (s) |
> | ---    | ---                | ---                                  |
> | NoPE   | 0.09               | 0.01                                 |
> | LaPE   | 0.10               | 164.83                               |
> | RWPE   | 0.10               | 0.18                                 |
> | LEAP-F | 0.58               | 11.57                                |
> | LEAP-L | 1.13               | 11.57                                |
>
> > The ablation study (Table 2 ) shows that 1-hop neighborhoods perform better than 2-hop or a 1,2-hop combination. This seems counter-intuitive to the MPNN notion that larger receptive fields are better. Why is 1-hop sufficient?
>
> Thanks for raising this relevant point\! Our current hypothesis is that this is due to a 10-dimensional final embedding. Hence, when using the $1,2$-hop neighborhoods, we project them into two 5-dimensional vectors, for a final 10-dimensional embedding. It indeed seems that the 1-hop neighborhood is already sufficient, but additionally adding the (most likely redundant) 2-hop neighborhood actually reduces expressivity, since we are projecting the ECT into two 5 dimensional vectors and thus the “useful” information now gets constrained to a 5-dimensional vector. The results we observed here are also consistent with the ablation studies performed in \[2\], pointing towards the fact **that in many real-world datasets a 1-hop view of the data is sufficient, provided that information can be combined effectively**. This is precisely an advantage of our method: **We represent the full graph via its $1$-hop neighborhood, providing an expressive mix between a local and global view on data.**
>
> ---
>
> \[1\] [Differentiable Euler Characteristic Transforms for Shape Classification](https://arxiv.org/abs/2310.07630)
>
> \[2\] [Diss-l-ECT: Dissecting Graph Data with Local Euler Characteristic Transforms](https://openreview.net/forum?id=GAmmzu6GYS)

---

> > ### Author Response · Authors · 2025-11-24
> >
> > We would like to thank the reviewer again for their efforts in reviewing our
> > work and very much appreciate the thorough review.
> > In our response we hope to have addressed the main concerns, namely providing **guidelines for
> > hyperparameters**, an **evaluation of the computational performance** of our
> > method and have shown that **LEAP works for very large graphs**. The revised
> > manuscript has been uploaded with the **changes highlighted in green**.
> >
> > If the reviewer believes we have adequately addressed their concerns we truly
> > appreciate it if they would reconsider their score and if they have any
> > questions we would be very happy to engage in further discussion.

---

> > > ### Author Response · Authors · 2025-11-28
> > >
> > > We thank the reviewer for their valuable comments. Since the rebuttal is coming to an end (and NeurIPS is about to start), we were wondering if there were any other points that we could clarify or experiments we could run. We believe that we have addressed your main concerns (in particular concerning hyperparameters and computational complexity) with our rebuttal answers and the revision; if so, we would kindly ask you to reconsider your overall score.

---

### Official Review · Reviewer_dZ9D · 2025-11-07

**Soundness:** 2
**Presentation:** 3
**Contribution:** 2
**Rating:** 6
**Confidence:** 4

**Summary:**

This paper introduces LEAP, an end-to-end trainable local structural positional encoding that integrates geometric and topological information via the local Euler Characteristic Transform (ℓ-ECT). LEAP computes node-level embeddings by extracting m-hop neighborhoods, normalizing local features, applying a differentiable ℓ-ECT, and projecting the resulting matrix into a compact representation through learnable modules (e.g., linear layers, 1D convolutions, or DeepSets). Two variants are proposed: LEAP-F, which uses fixed ECT directions, and LEAP-L, which learns them during training. Extensive experiments on both synthetic and real-world datasets, using three GNN architectures, confirm that LEAP substantially enhances graph representation learning.

**Strengths:**

S1: The paper proposes an end-to-end trainable framework for encoding positional information, eliminating the need for static or precomputed preprocessing.

S2: The experimental setup is well-structured, evaluating three mainstream GNN architectures across multiple datasets, including both synthetic and real-world benchmarks.

S3: Ablation studies demonstrate LEAP’s robustness to key hyperparameters and locality configurations. Moreover, the five projection strategies showcase LEAP’s adaptability to diverse model architectures and task requirements.

**Weaknesses:**

W1: Although the paper discusses LEAP’s robustness to hyperparameters (e.g., number of directions, thresholds), it omits computational efficiency analysis.

W2: Since no single projection strategy yields the best results across all tasks, do the authors have heuristic or data-driven guidelines for choosing among them

W3: Theoretical analysis and justification are insufficient, which is critical for this method-oriented paper.

**Questions:**

Q1: On the HIV dataset, the locally encoded LEAP underperforms the globally informed LaPE. Could the authors elaborate on why global structural information is crucial for this task?

Q2: The injectivity of the Euler Characteristic Transform underpins its discriminative power, yet LEAP employs a differentiable local approximation. Do the authors have theoretical or empirical evidence that this approximation preserves key topological information compared with the exact, global ECT?

Q3: Is LEAP applicable to geometric graphs? If so, how does it leverage spatial coordinates or geometric relationships within the ℓ-ECT framework?

---

> ### Author Response · Authors · 2025-11-18
> **Response 1/2**
>
> We thank the reviewer very much for their positive review and acknowledgement of our work. In our work, we aimed for a **strong experimental section with sufficient ablations and we appreciate the reviewer for recognizing that**.
>
> Please find our responses below.
>
> > W1: Although the paper discusses LEAP’s robustness to hyperparameters (e.g., number of directions, thresholds), it omits computational efficiency analysis.
>
> We thank the reviewer for this suggestion. **In our updated manuscript we have added a complexity analysis (Section 3.3)**. As the computational performance is important to us and relevant for applicability, we also provide training times for **very large graphs below** and will ensure the final version of the manuscript reflects this. In general, computing the ECT can be done efficiently in parallel. Notice that positional encoding encodings like LaPE require eigendecomposition, which becomes prohibitive for larger graphs; we already observe this in the `Roman-Empire` graphs (see below). As the table of timings below shows, our implementation remains **practically usable**, and we believe that further improvements to our implementation (like precomputing neighborhoods) are possible.
>
> Letter-High:
> | Method | Average Training (s) | Average Time/Epoch (s) | Average Epochs |
> | ---    | ---                  | ---                    | ---            |
> | NoPE   | 44.60            | 0.51               | 86.4           |
> | LaPE   | 60.35            | 0.64               | 93.0           |
> | RWPE   | 35.77            | 0.50               | 71.2           |
> | LEAP-L | 32.51            | 0.93               | 34.8           |
> | LEAP-F | 30.08            | 0.84               | 35.8           |
>
> Roman-Empire:
> | Method | Avg Epoch Time (s) | Avg pre-processing time (s) |
> | ---    | ---                | ---                                  |
> | NoPE   | 0.09               | 0.01                                 |
> | LaPE   | 0.10               | 164.83                               |
> | RWPE   | 0.10               | 0.18                                 |
> | LEAP-F | 0.58               | 11.57                                |
> | LEAP-L | 1.13               | 11.57                                |
>
> > W2: Since no single projection strategy yields the best results across all tasks, do the authors have heuristic or data-driven guidelines for choosing among them.
>
> We thank the reviewer for raising this point as it will most certainly improve the adoption of our method in applications. According to our experiments, choosing $1$-hop neighborhoods and $16$ directions with a resolution of $16$ (thus a $16\\times 16$ image) provides a sufficiently strong starting point. As a projection strategy, based on our experimental results,we suggest starting with an attention-based projection, switching to a linear projection in case of overfitting.
>
> When the model has the tendency to over- or underfit, decreasing (resp. increasing) the number of directions and resolution is a suitable option, as well. In choosing the right scale for the sigmoid approximation in the DECT, we follow \[1\] and the settings therein. In a recent paper \[2\], the authors perform an ablation with respect to the scale and derive the principle that setting the scale $¼$ of the resolution yields the best result. **We will provide additional guidelines on choosing these hyperparameters for the final version of our manuscript**.
>
> > W3: Theoretical analysis and justification are insufficient, which is critical for this method-oriented paper.
>
> The theoretical underpinning of method-oriented papers like ours is indeed crucial. Since our paper emphasizes method development, we rely on the already-existing theoretical results \[2, 3\] that discuss expressivity. Notably, the authors of \[3\] show that the local ECT **is at least as expressive as message passing**. We now **added a discussion of these properties to our revised manuscript**, and will extend this discussion in the appendix for the final version.
>
> Please find the responses to your questions below.
>
> —
>
> \[1\] [Differentiable Euler Characteristic Transforms for Shape Classification](https://arxiv.org/abs/2310.07630)
>
> \[2\] [Point Cloud Synthesis Using Inner Product Transforms](https://arxiv.org/abs/2410.18987)
>
> \[3\] [Diss-l-ECT: Dissecting Graph Data with Local Euler Characteristic Transforms](https://arxiv.org/abs/2410.02622)

---

> ### Author Response · Authors · 2025-11-18
> **Response 2/2**
>
> > Q1: On the HIV dataset, the locally encoded LEAP underperforms the globally informed LaPE. Could the authors elaborate on why global structural information is crucial for this task?
>
> We thank the reviewer for this insightful feedback. In \[1\] the DECT is used as a *global* graph representation, whereas we use it to provide a *local* node representation. To complement the local nature of the $\\ell$-ECT, it is possible to add a global representation as input of the model. For instance, one could inject the projection of the global ECT into the final classification layer for further improving expressivity. We believe that the advantage of our method lies in its flexibility, making it possible to build architectures that accommodate different tasks.
>
> In response to reviewer `8cdq`, we have **added an additional experiment for the HIV dataset, where LEAP is complemented with the LaPE, showing strong improvements over the baseline**.
>
> | Model         | Accuracy     |
> | ---            |  --- |
> | NoPE  | 69.48 $\pm$ 0.60 |
> | LaPE  | 72.94 $\pm$ 2.09 |
> | RWPE   | 70.62 $\pm$ 1.68 |
> | LEAP-F        | 71.79 $\pm$ 2.62 |
> | LEAP-L        | 71.38 $\pm$ 1.72 |
> | LEAP-F & LaPE | 74.14 $\pm$ 2.22 |
> | LEAP-L & LaPE | 73.59 $\pm$ 1.57 |
>
> > Q2: The injectivity of the Euler Characteristic Transform underpins its discriminative power, yet LEAP employs a differentiable local approximation. Do the authors have theoretical or empirical evidence that this approximation preserves key topological information compared with the exact, global ECT?
>
> The injectivity of the ECT is not only guaranteed in the theoretical sense, but also in the practical sense. In \[1, 2\] for instance, the authors show that one can actually **invert the ECT for point clouds and do this in particular with the differentiable ECT**. That said, the inversion of  ECT for graphs remains an active area of research and is to date **not yet fully solved**. We will extend the discussion of this in our final manuscript.
>
> > Q3: Is LEAP applicable to geometric graphs? If so, how does it leverage spatial coordinates or geometric relationships within the ℓ-ECT framework?
>
> Indeed, the core assumption of our framework is that the graph **is a geometric graph with node features**. We apologize for phrasing this confusingly and have since **updated the introduction and discussion** in our revision, stating clearly that the node features, together with the direction vectors, are used to “view” and analyze a graph from multiple perspectives.
>
> —
>
> \[1\] [Differentiable Euler Characteristic Transforms for Shape Classification](https://arxiv.org/abs/2310.07630)
>
> \[2\] [Point Cloud Synthesis Using Inner Product Transforms](https://arxiv.org/abs/2410.18987)
>
> \[3\] [Diss-l-ECT: Dissecting Graph Data with Local Euler Characteristic Transforms](https://arxiv.org/abs/2410.02622)

---

> > ### Author Response · Authors · 2025-11-24
> >
> > We appreciate the great feedback and support from the reviewer, which has been very
> > valuable to both the clarity as well as contribution of our work.
> >
> > In particular we have implemented the following changes:
> > - We provide additional **guidelines for hyperparameters**.
> > - We provide more details on **choosing the projection strategy**.
> > - We expand the **theoretical motivation** and explain why additional theory is out
> > of scope for this work.
> >
> > In our response we have hopefully addressed the concerns and we have updated the
> > manuscript (changes highlighted in green).
> >
> > If the reviewer believes we have adequately addressed their concens we would
> > very much appreciate if they would reconsider their score and we would be very
> > happy to engage in further discussion.

---

> > > ### Author Response · Authors · 2025-11-28
> > >
> > > We thank the reviewer for their valuable comments. Since the rebuttal is coming to an end (and NeurIPS is about to start), we were wondering if there were any other points that we could clarify or experiments we could run. We believe that we have addressed your main concerns with our rebuttal answers and the revision; if so, we would kindly ask you to reconsider your overall score.

---

### Author Response · Authors · 2025-12-03
**Summary of the rebuttal**

Dear AC, dear reviewers, dear all,

Thanks for engaging with our work, in particular in light of the data leak. For the benefit of this discussion, we briefly summarize our rebuttal here:

- Reviewer `dZ9D` (initial rating 6): Raised questions about choosing hyperparameters and requested more information on complexity and theoretical justifications.

  **Our Answer**: We provided such theoretical justifications and heuristics for choosing hyperparameters in our revised manuscript. We also commented on computational complexity with a table of timings, showing the *scalability* of our method. We also provided additional experiments concerning the global versus local behavior of our method, showing that LEAP (a _local_ encoding), in combination with _global_ encodings

- Reviewer `6i1u` (initial rating 4): Raised questions about choosing hyperparameters and the utility of LEAP for non-geometric graphs as well as the scalability.

  **Our Answer**: We provided a discussion on choosing hyperparameters in our revision. We also provided comments on the general utility of LEAP for non-geometric graphs (referring to existing sections in the manuscript, but also providing new details). Finally, we added a table of timings, showing the *scalability* of our method.

- Reviewer `8cdq` (initial rating 4): Raised questions about the theoretical justification. Asked whether LEAP can be combined with global positional encodings and requested details on computational complexity.

  **Our Answer**: We provided additional context for the theoretical motivations, evaluated LEAP with a *global positional encoding* (showing improvements in predictive performance) and added a thorough analysis of the runtime and complexity of our method, finding that it remains highly scalable.

- Reviewer `f41T` (initial rating 6, updated to **8** before the reset): Requested additional comparisons and suggested improvements to the text; in particular, the reviewer suggested that we can word some claims more strongly.

  **Our Answer**: We revised the manuscript accordingly and added new experiments with additional GNNs and with global positional encodings, showing improved predictive performance.

---

Except for reviewer `f41T` (who **raised their score to an 8**), none of the reviewers unfortunately responded before the discussion was closed on Nov 28/29 due to the data leakage. **We believe to have addressed all issues raised by the reviewers** and strengthened our manuscript accordingly. We hope that the non-responsiveness will not be detrimental in the final assessment of our work.

Sincere thanks to everyone for helping us make our work better. 🙌

---

### Meta-Review · Area_Chair_Gg2o · 2026-01-07

**Summary:**

The paper presents a new strategy for Graph Positional Encoding, based on a differentiable Euler Characteristic Transform. Reviewers generally comment that the paper is clear and well-written, and that the empirical results are very good. Yet, concerns were raised on three main axis: local vs global expressive power, computational tractability and comparisons with other GNN architectures. I believe that those points have been succesfully addressed by authors during the rebuttal phase. Despite a somehow relatively low score, I recommend this paper for acceptance in ICLR2026 program.

**Reviewer Concerns:**

Concerns addressed by authors:
- computational complexity / tractability: despite globally higher computation times, authors have shown that their method remains tractable on large graphs.
- Locality vs Globality: authors show that while their encoding is definitely local, adding a global descriptor improves significantly the results
- comparisons with other architectures (e.g. GIN): additional baselines were added to the results, which do not change the conclusions.

Concern still pending: relatively moderated originality of the method, which is counterbalanced by strong empirical results.

**Reviewer Scores:**

f41T would have raised his score to 8. The other reviewers did not engage strongly in the discussion phase.

---

### Decision · Program_Chairs · 2026-01-26

Accept (Poster)